# BADiff: Bandwidth Adaptive Diffusion Model

**Xi Zhang**[1]    **Hanwei Zhu**[1]✉    **Yan Zhong**[1]    **Jiamang Wang**[2]    **Weisi Lin**[1]✉

[1]Nanyang Technological University    [2]Alibaba Group

{xi.zhang, hanwei.zhu, wslin}@ntu.edu.sg

## Abstract

In this work, we propose a novel framework to enable diffusion models to adapt their generation quality based on real-time network bandwidth constraints. Traditional diffusion models produce high-fidelity images by performing a fixed number of denoising steps, regardless of downstream transmission limitations. However, in practical cloud-to-device scenarios, limited bandwidth often necessitates heavy compression, leading to loss of fine textures and wasted computation. To address this, we introduce a joint end-to-end training strategy where the diffusion model is conditioned on a target quality level derived from the available bandwidth. During training, the model learns to adaptively modulate the denoising process, enabling early-stop sampling that maintains perceptual quality appropriate to the target transmission condition. Our method requires minimal architectural changes and leverages a lightweight quality embedding to guide the denoising trajectory. Experimental results demonstrate that our approach significantly improves the visual fidelity of bandwidth-adapted generations compared to naive early-stopping, offering a promising solution for efficient image delivery in bandwidth-constrained environments. Code is available at: https://github.com/xzhang9308/BADiff.

## 1 Introduction

Diffusion models [13, 43, 15, 37, 40] have recently demonstrated remarkable capabilities in synthesizing high-quality images, significantly surpassing previous generative approaches such as GANs [10, 36, 2, 19] and VAEs [21, 38, 12, 47]. Despite their impressive fidelity, deploying diffusion models in realistic cloud-to-user applications introduces a fundamental bottleneck: transmission bandwidth. In conventional scenarios, generated images undergo aggressive lossy compression [45, 46, 4, 44, 1, 5, 34, 32, 58, 55] before transmission to accommodate limited bandwidth. This cascaded approach—high-quality image generation followed by subsequent compression—not only incurs redundant computational overhead but also significantly degrades perceptual quality, as the compression process often erases the intricate textures and fine details [24, 8, 56, 57, 7, 25, 16, 48] carefully constructed by the diffusion model.

This motivates a critical question: *Can we directly integrate bandwidth-awareness into the diffusion generation process, avoiding the inefficiency and perceptual quality loss associated with post-generation compression?* Diffusion models inherently provide a natural mechanism for addressing this challenge. During sampling, these models progressively refine coarse structures into detailed, realistic textures through iterative denoising steps. Thus, intuitively, terminating the diffusion process early results in simpler, lower-detail images suitable for constrained bandwidth scenarios. However, naively reducing the number of diffusion steps typically produces suboptimal visual quality, as models trained for complete denoising trajectories are not optimized for early termination, leading to visual artifacts and poor perceptual coherence.

---

✉ Corresponding authors.

39th Conference on Neural Information Processing Systems (NeurIPS 2025).

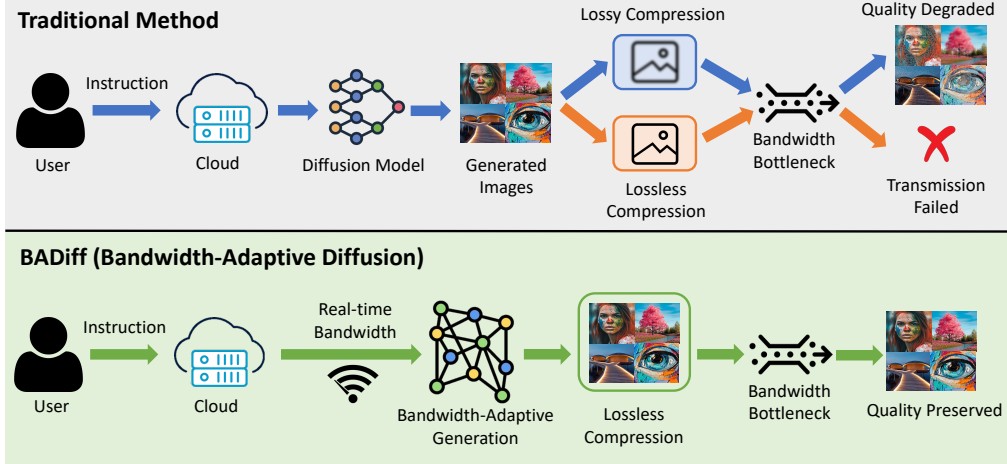

Figure 1: Comparison of traditional diffusion + compression pipeline (top) and the proposed BADiff framework (bottom). BADiff directly generates entropy-constrained images suitable for bandwidth-limited transmission, avoiding quality degradation in post-generation compression.

To effectively address this issue, we propose the ***Bandwidth-Adaptive Diffusion Model (BADiff)***, a novel diffusion model explicitly conditioned on target bandwidth constraints, formulated as entropy targets. By embedding target entropy as an explicit conditioning input, our approach enables the diffusion model to adaptively modulate its denoising behavior. During training, the model is exposed to variable entropy constraints, allowing it to produce perceptually pleasing images even under reduced-step sampling. Furthermore, an entropy regularization loss ensures the generated images adhere closely to the bandwidth constraints, obviating the need for aggressive post-hoc compression. As illustrated in Figure 1, the conventional diffusion pipeline relies on lossy compression to meet bandwidth constraints, often degrading visual quality, whereas our proposed BADiff framework directly generates entropy-constrained outputs suitable for transmission without compromising perceptual fidelity.

Our proposed BADiff framework provides several advantages over existing cascaded and naive early-stopping approaches. First, it significantly reduces computational overhead by adaptively terminating sampling, ensuring efficient inference. Second, by directly generating images meeting bandwidth constraints, BADiff avoids compression-induced artifacts, thus preserving perceptual quality. Third, our framework provides fine-grained, dynamic control over image quality based on real-time bandwidth conditions, enhancing deployment flexibility in practical cloud-to-user applications.

We validate the effectiveness of BADiff through extensive experiments comparing our method to strong baselines, including standard diffusion models with post-generation compression and naive early-stopping approaches. Our results demonstrate that BADiff consistently achieves superior trade-offs among perceptual quality, computational efficiency, and bandwidth efficiency, underscoring the advantage of integrating bandwidth-awareness directly into the generative modeling process.

Our main contributions can be summarized as follows:

- We introduce BADiff, the first bandwidth-adaptive diffusion model explicitly conditioned on target entropy constraints, directly addressing image synthesis for bandwidth-constrained transmission.

- We propose an entropy conditioning mechanism integrated into diffusion models, coupled with an entropy regularization loss, allowing adaptive and efficient generation under diverse bandwidth constraints.

- We develop an adaptive sampling policy that dynamically determines optimal sampling termination, significantly reducing computational cost while preserving image quality.

- Through extensive evaluations, we demonstrate BADiff's superior performance in perceptual quality, computational efficiency, and adherence to bandwidth constraints compared to conventional cascaded diffusion + compression pipelines.

## 2 Related Work

### 2.1 Diffusion Models

Diffusion models have recently gained significant attention due to their remarkable ability to generate high-quality images, surpassing traditional generative models such as GANs [10] and VAEs [21]. The foundational work of Ho et al. [13] introduced Denoising Diffusion Probabilistic Models (DDPMs), formalizing diffusion models as a parameterized Markov chain trained by variational inference to invert a gradual noising process. Song et al. [43] further generalized the framework through Score-based Generative Models (SGMs), which unify diffusion models and score-matching approaches under a continuous-time stochastic differential equation (SDE) framework. Recent works have extended diffusion models to various tasks beyond image synthesis, including video generation [15], text-to-image generation [37, 40], and 3D synthesis [35].

### 2.2 Accelerated Sampling of Diffusion Models

Despite their high-quality outputs, diffusion models are computationally expensive due to the iterative sampling procedure required during generation. To mitigate this issue, substantial efforts have been made toward accelerating diffusion model sampling. DDIM [42] introduced deterministic sampling methods enabling fewer inference steps, significantly reducing computation. Further advances, including DPM-Solver [29] and FastDPM [22], have employed numerical methods inspired by ordinary differential equations (ODEs) to shorten sampling times substantially while preserving generation quality. PNDM [28] introduces a pseudo–numerical solver that treats the reverse diffusion ODE with high-order Runge–Kutta–style updates, enabling high-fidelity image generation in as few as four forward passes. Knowledge distillation based methods [41, 31] reduce sampling steps by distilling knowledge from slower teacher models into faster student models. Alternative approaches involve adaptive step size selection [17] or latent space compression [39] to reduce computational overhead. AutoDiffusion [26] further accelerates sampling via non-uniform step skipping.

Unlike prior methods that optimize scheduling or model architecture, DDSM [49] introduces dynamic U-Net pruning to minimize redundant computations at each step. This approach is complementary to existing acceleration techniques. Related works include OMS-DPM [27], which optimizes model scheduling via predictor-based algorithms, and Spectral Diffusion [50], which employs dynamic gating for efficiency. While eDiff-I [3] uses multiple fixed-size expert models, StepSaver [52] proposes a step predictor to determine the minimal denoising steps required for high-quality generation, further improving efficiency. Moreover, adaptive methods such as AdaDiff [53] dynamically adjust inference trajectories, further enhancing efficiency by selectively allocating computational resources during sampling based on intermediate outputs.

### 2.3 Conditional and Controllable Diffusion Models

Diffusion models inherently provide powerful frameworks for conditional and controllable generation. Classifier-guided diffusion [9] utilizes gradients from auxiliary classifiers to steer the generative process toward desired attributes. However, training classifiers separately is often cumbersome and computationally expensive. Classifier-free guidance [14] resolves this issue by training diffusion models on conditional and unconditional inputs simultaneously, enabling flexible attribute control without additional classifiers. Latent diffusion models (LDMs) [39] further enhance controllability and computational efficiency by conditioning generation in latent spaces. Recent approaches like RePaint [30] have explored numerical control, such as region-based conditioning, to edit images interactively, although they do not directly address bandwidth constraints or entropy-aware generation. Our proposed BADiff model differs by conditioning generation directly on entropy constraints, enabling explicit control over the generated image's compressibility and perceptual quality.

## 3 BADiff

We propose **BADiff (Bandwidth-Adaptive Diffusion)**, a conditional diffusion framework that integrates bandwidth constraints, formulated as target entropy values, into the diffusion sampling (see Fig. 2). BADiff aims to dynamically modulate generation to satisfy entropy constraints during generation, thus eliminating the need for post-hoc compression while saving computational cost.

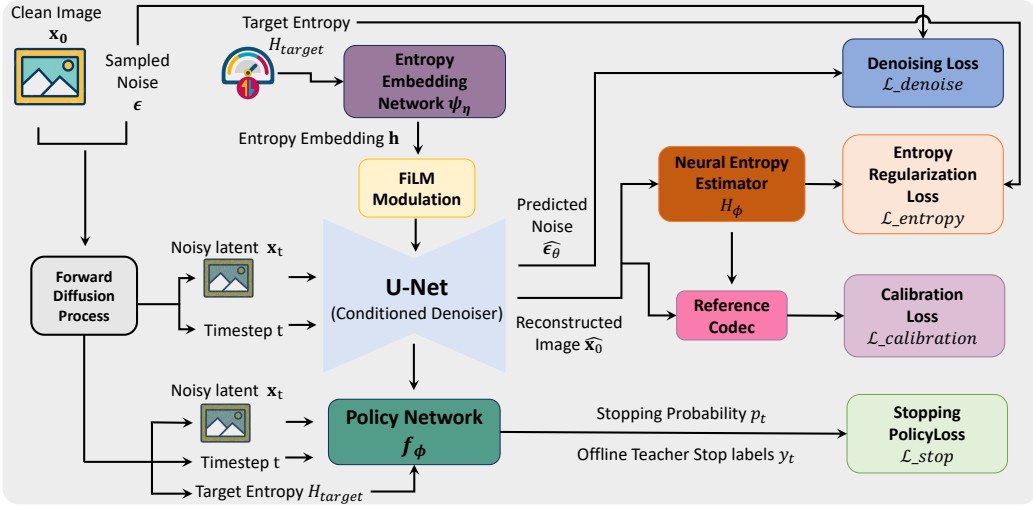

Figure 2: **Overview of the BADiff training framework.** The proposed framework jointly optimizes image generation quality and bandwidth adaptability. The total training objective integrates four complementary losses: (1) the standard Denoising Loss $\mathcal{L}_{\text{denoise}}$ for reconstruction; (2) an Entropy Regularization Loss $\mathcal{L}_{\text{entropy}}$ enforced by a differentiable Neural Entropy Estimator $H_\phi$ to ensure the budget is met; (3) a Calibration Loss $\mathcal{L}_{\text{calibration}}$ that aligns the estimator with a Reference Codec; and (4) a Stopping Policy Loss $\mathcal{L}_{\text{stop}}$ to train the lightweight Policy Network $f_\phi$ for adaptive early exiting.

## 3.1 Background: Diffusion Models

Diffusion models generate data by reversing a forward process that gradually adds Gaussian noise. Formally, given a clean sample $\mathbf{x}_0$, the noisy latent $\mathbf{x}_t$ at timestep $t$ can be sampled directly via the reparameterization trick:

$$\mathbf{x}_t = \sqrt{\bar{\alpha}_t}\,\mathbf{x}_0 + \sqrt{1 - \bar{\alpha}_t}\,\boldsymbol{\epsilon}, \quad \boldsymbol{\epsilon} \sim \mathcal{N}(0, \mathbf{I}), \tag{1}$$

where $\bar{\alpha}_t$ is derived from a fixed noise schedule. The generative process relies on a neural network $\boldsymbol{\epsilon}_\theta(\mathbf{x}_t, t)$ trained to predict the added noise $\boldsymbol{\epsilon}$. The standard training objective minimizes the simple mean-squared error:

$$\mathcal{L}_{\text{denoise}} = \mathbb{E}_{\mathbf{x}_0, t, \boldsymbol{\epsilon}} \left[ \|\boldsymbol{\epsilon} - \boldsymbol{\epsilon}_\theta(\mathbf{x}_t, t)\|_2^2 \right]. \tag{2}$$

By iteratively removing the predicted noise starting from $\mathbf{x}_T \sim \mathcal{N}(0, \mathbf{I})$, the model reconstructs the data distribution.

## 3.2 Entropy–Conditioned Diffusion Model

In a standard DDPM, the reverse process refines noisy latents $\mathbf{x}_t$ into a clean image $\mathbf{x}_0$ through a Markov chain with Gaussian transitions $p_\theta(\mathbf{x}_{t-1} \mid \mathbf{x}_t) = \mathcal{N}\big(\mathbf{x}_{t-1}; \boldsymbol{\mu}_\theta(\mathbf{x}_t, t), \boldsymbol{\Sigma}_\theta(\mathbf{x}_t, t)\big)$. Such models ignore external resource constraints (e. g. bandwidth). BADiff enforces the constraint by conditioning every reverse step on a target entropy budget $H_{\text{target}} \in \mathbb{R}_{>0}$.

Specifically, we extend the reverse kernel to

$$p_\theta(\mathbf{x}_{t-1} \mid \mathbf{x}_t, H_{\text{target}}) = \mathcal{N}\Big(\mathbf{x}_{t-1}; \boldsymbol{\mu}_\theta(\mathbf{x}_t, t, H_{\text{target}}), \boldsymbol{\Sigma}_\theta(\mathbf{x}_t, t, H_{\text{target}})\Big), \tag{3}$$

so that the predicted noise (or velocity) becomes $\hat{\boldsymbol{\epsilon}}_\theta = \boldsymbol{\epsilon}_\theta(\mathbf{x}_t, t, H_{\text{target}})$.

A single scalar is not expressive enough for deep conditioning, therefore we map $H_{\text{target}}$ into a $d$–dimensional vector via a learned *entropy embedding network*

$$\mathbf{h} = \psi_\eta\big(H_{\text{target}}\big) \in \mathbb{R}^d, \tag{4}$$

where $\psi_\eta : \mathbb{R} \to \mathbb{R}^d$ is an MLP with parameters $\eta$. Throughout the paper we fix $d = 128$.

Let $\mathbf{g}(t)$ be the usual sinusoidal timestep embedding. For every residual block $l$ of the UNet we form a hybrid modulation

$$\mathbf{g}_l(t, H_{\text{target}}) = \mathbf{g}(t) + \mathbf{W}^{(l)}\mathbf{h}, \tag{5}$$

and add $\mathbf{g}_l$ to the block's activation just before the first convolution (equivalent to additive FiLM). Here $\mathbf{W}^{(l)} \in \mathbb{R}^{c_l \times d}$ is learned per-block and $c_l$ is the channel width. Consequently every output of the denoiser, $\epsilon_\theta(\mathbf{x}_t, t, H_{\text{target}})$, is explicitly conditioned on the entropy budget.

This design keeps the overhead negligible ($< 0.1\%$ additional parameters) while giving the network a continuous control "dial" over the amount of detail it should recover, enabling BADiff to generate images whose bit-rate naturally matches the specified bandwidth.

### 3.3 Entropy Regularization Loss

Conditioning the reverse process on an entropy budget is *necessary but not sufficient*: the model could still output images whose empirical entropy exceeds the target. To make the constraint *active* during learning we attach an explicit penalty that is *differentiable* w.r.t. both the image and the network parameters.

Let $\hat{\mathbf{x}}_0 = g_\theta(\mathbf{x}_t, t, H_{\text{target}})$ be the reconstructed clean sample predicted at time–step $t$. We introduce

$$\mathcal{L}_{\text{entropy}} = \max\left(0, H_\phi(\hat{\mathbf{x}}_0) - H_{\text{target}}\right), \tag{6}$$

where $H_\phi(\cdot)$ is a *differentiable* neural entropy estimator parametrized by $\phi$ and jointly optimized with $\theta$. The hinge form ensures that no gradient flows once the sample entropy is already below the budget, so the optimizer focuses on over-budget cases.

**Differentiable neural entropy predictor.** In learned image compression [5, 34, 33], the entropy model predicts a *pixel–wise conditional distribution* $p_\phi(x_u \mid \mathbf{c}_u)$ given causal context $\mathbf{c}_u$ (e.g. neighbouring pixels or a hyper-prior) and converts it to code-length via $-\log_2 p_\phi$. We adopt the same principle for BADiff.

For each spatial position $u \in \Omega$ the entropy network $E_\phi$ outputs continuous parameters $\boldsymbol{\theta}_u = (\boldsymbol{\mu}_u, \boldsymbol{\sigma}_u)$ of a *discretized logistic* distribution [5]:

$$p_\phi(x_u \mid \mathbf{c}_u) = \mathcal{DL}(x_u; \mu = \mu_u, \sigma = \sigma_u). \tag{7}$$

The expected code-length (bits-per-pixel) for an image $\mathbf{x}$ is therefore

$$H_\phi(\mathbf{x}) = -\frac{1}{|\Omega|} \sum_{u \in \Omega} \log_2 p_\phi(x_u \mid \mathbf{c}_u), \tag{8}$$

which is fully differentiable w.r.t. both $\boldsymbol{\theta}_u$ and the upstream activations that determine $\mathbf{c}_u$; hence gradients flow into the UNet.

**Context extraction.** We follow [5, 34] and construct $\mathbf{c}_u$ from two sources: (i) a *hyper-prior* $\mathbf{z}$ predicted by a lightweight conv-net over $\mathbf{x}$, and (ii) an auto-regressive causal context of previously decoded pixels (realised as masked convolutions).

**Gradient properties.** Because the discrete logistic PMF is analytically differentiable w.r.t. $\boldsymbol{\mu}_u$ and $\boldsymbol{\sigma}_u$, Eq. (8) supplies exact gradients: $\nabla_{\boldsymbol{\theta}_u} H_\phi(\mathbf{x}) = -\frac{1}{\ln 2} \nabla_{\boldsymbol{\theta}_u} \log p_\phi(x_u \mid \mathbf{c}_u)$. Hence the entropy constraint is enforced *end-to-end*, unlike histogram-based surrogates that require straight-through tricks.

**Self-supervised calibration of $E_\phi$.** Although $E_\phi$ is trained jointly via the hinge loss $\mathcal{L}_{\text{ent}}$, it benefits from an auxiliary signal that anchors its probabilities to a *known* codec. To this end we derive pixel-wise targets $q_u(k)$ from a reference end-to-end optimized image codec. We then minimize the spatially averaged cross–entropy

$$\mathcal{L}_{\text{calibration}} = \frac{1}{|\Omega|} \sum_{u \in \Omega} \sum_{k=1}^{K} q_u(k) \left[-\log_2 p_\phi(k \mid \mathbf{c}_u)\right], \tag{9}$$

which is equivalent (up to a constant) to the KL-divergence $D_{\text{KL}}(q_u \parallel p_\phi)$. This term *calibrates* the logits toward realistic code-lengths without over-regularizing.

Because $H_\phi$ is differentiable, the model learns a *direct mapping* from an entropy budget to the statistics of its output, which empirically accelerates convergence and yields tighter adherence to the target bandwidth than heuristic early stopping.

---

We use "entropy" as a shorthand for the *expected code-length* (bits-per-pixel) after an entropy–coding stage.

## 3.4 Adaptive Sampling Policy

Let $\tau \in \{1, \ldots, T\}$ denote the (random) stopping time at which sampling terminates and the current latent $\mathbf{x}_\tau$ is decoded to the final image $\hat{\mathbf{x}}_0$. Ideally we would like to choose $\tau$ so as to minimize the *total cost*

$$\mathcal{C}(\tau) \;=\; \underbrace{\mathcal{E}(\hat{\mathbf{x}}_0)}_{\text{entropy}} \;+\; \underbrace{\beta\,\mathcal{D}(\hat{\mathbf{x}}_0, \mathbf{x}_{\text{ref}})}_{\text{distortion}} \;+\; \underbrace{\gamma\,\tau}_{\text{compute}}, \tag{10}$$

where $\mathcal{E}$ is the entropy predictor $H_\phi(\cdot)$, $\mathcal{D}$ is a perceptual distortion (e.g. LPIPS to the reference $\mathbf{x}_{\text{ref}}$), and $\beta, \gamma > 0$ weigh quality vs. runtime. Brute–force evaluation at all $t$ is impossible during inference, so we approximate $\tau$ with a lightweight classifier that decides *on the fly* whether to proceed.

**Policy network.** We introduce a small MLP-based policy network $f_\phi : \mathbb{R}^d \times \mathbb{N} \times \mathbb{R} \to [0, 1]$ that outputs the *stop - probability*

$$p_t \;=\; f_\phi\big(\mathbf{z}_t,\, t,\, H_{\text{target}}\big), \qquad \mathbf{z}_t = \tfrac{1}{hw}\sum_{u \in \Omega}\mathbf{x}_t[u] \in \mathbb{R}^d, \tag{11}$$

where $\mathbf{z}_t$ is a spatial mean-pooled latent feature ($h, w$ are height/width, $d$ channels). Sampling continues iff a Bernoulli draw $b_t \sim \text{Bernoulli}(1 - p_t)$ returns 1. Thus the stopping time is $\tau = \min\{\, t \mid b_t = 0\} \vee 1$.

**Supervised self–distillation.** We generate *teacher* stop–labels offline: run a long-step sampler, measure the cost $\mathcal{C}(t)$ at each step and set

$$y_t \;=\; \mathbb{1}\big[\mathcal{C}(t) \leq \min_{s \geq t}\mathcal{C}(s)\big]. \tag{12}$$

Hence $y_t = 1$ iff step $t$ is sufficient; earlier steps are labelled $0$. The policy is trained jointly with BADiff via

$$\mathcal{L}_{\text{stop}} \;=\; \mathbb{E}_t\big[\text{BCE}(y_t,\, p_t)\big], \tag{13}$$

where BCE is binary cross-entropy. Gradients back-propagate through $p_t$ but *not* through the discrete Bernoulli sample (stop/no-stop), ensuring stable training.

**Inference procedure.** During sampling we evaluate $p_t$ at each step: if $p_t \geq \tau_{\text{th}}$ ($\tau_{\text{th}} = 0.5$ by default) we terminate and decode $\mathbf{x}_t$; otherwise we proceed to $t-1$. Because the policy is only a several layer MLP over $\mathbf{z}_t$ and $(t, H_{\text{target}})$, the additional runtime overhead is negligible ($< 0.3$ ms per step on RTX 4090). Empirically (§4.3) BADiff stops about **50% earlier** on low-bandwidth budgets while keeping LPIPS and FID constant, demonstrating the benefit of the adaptive sampling policy.

## 3.5 Training and Sampling

The full BADiff objective merges **four** complementary loss terms: *(i)* the standard denoising loss from DDPM, *(ii)* the entropy hinge that enforces the bandwidth budget, *(iii)* a calibration loss that aligns the learned entropy model with a reference codec, and *(iv)* a stopping-policy loss that teaches the lightweight classifier when to terminate sampling. Formally,

$$\mathcal{L} = \underbrace{\mathbb{E}_{\mathbf{x}_0, t, \boldsymbol{\epsilon}}\Big[\big\|\boldsymbol{\epsilon} - \boldsymbol{\epsilon}_\theta(\mathbf{x}_t, t, H_{\text{target}})\big\|_2^2\Big]}_{\mathcal{L}_{\text{denoise}}} + \lambda_{\text{ent}}\underbrace{\max\big(0, H_\phi(\hat{\mathbf{x}}_0) - H_{\text{target}}\big)}_{\mathcal{L}_{\text{entropy}}}$$

$$+ \;\lambda_{\text{cal}}\underbrace{\frac{1}{|\Omega|}\sum_{u \in \Omega}D_{\text{KL}}\big(q_u \,\|\, p_\phi(\cdot \,|\, \mathbf{c}_u)\big)}_{\mathcal{L}_{\text{calibration}}} + \lambda_{\text{stop}}\underbrace{\mathbb{E}_t\big[\text{BCE}\big(y_t,\, f_\phi(\mathbf{x}_t, t, H_{\text{target}})\big)\big]}_{\mathcal{L}_{\text{stop}}}. \tag{14}$$

During training we randomly draw $H_{\text{target}} \sim \mathcal{U}(H_{\min}, H_{\max})$ to expose the network to a broad range of bandwidth budgets, allowing it to generalize to unseen conditions. At test time a user-specified bitrate is converted to an entropy budget $H_{\text{target}}$. Conditioned on this value, BADiff starts from a Gaussian latent and runs the reverse process. After each step the policy network $f_\phi(\mathbf{x}_t, t, H_{\text{target}})$ outputs a stop-probability; sampling terminates as soon as the probability exceeds a threshold $\tau_{\text{th}}$ (0.5 by default).

**Algorithm 1 BADiff Training**

1: **repeat**
2: $\quad \mathbf{x}_0 \sim q(\mathbf{x}_0)$
3: $\quad H_{\text{target}} \sim \mathcal{U}(H_{\min}, H_{\max})$
4: $\quad t \sim \mathcal{U}\{1, \dots, T\}, \ \boldsymbol{\epsilon} \sim \mathcal{N}(\mathbf{0}, \mathbf{I})$
5: $\quad \mathbf{x}_t = \sqrt{\bar{\alpha}_t}\mathbf{x}_0 + \sqrt{1 - \bar{\alpha}_t}\,\boldsymbol{\epsilon}$
6: $\quad \hat{\boldsymbol{\epsilon}}_\theta \leftarrow \boldsymbol{\epsilon}_\theta(\mathbf{x}_t, t, H_{\text{target}})$
7: $\quad \hat{\mathbf{x}}_0 \leftarrow g_\theta(\mathbf{x}_t, t, H_{\text{target}})$
8: $\quad \mathcal{L}_{\text{DEN}} = \left\| \boldsymbol{\epsilon} - \hat{\boldsymbol{\epsilon}}_\theta \right\|_2^2$
9: $\quad \mathcal{L}_{\text{ENT}} = \max\!\left(0,\ H_\phi(\hat{\mathbf{x}}_0) - H_{\text{target}}\right)$
10: $\quad \mathcal{L}_{\text{CAL}} = \frac{1}{|\Omega|}\sum_{u \in \Omega} D_{\text{KL}}\!\left(q_u \,\|\, p_\phi(\cdot \,|\, \mathbf{c}_u)\right)$
11: $\quad$ Generate teacher label $y_t$
12: $\quad p_t \leftarrow f_\phi(\mathbf{x}_t, t, H_{\text{target}})$
13: $\quad \mathcal{L}_{\text{STOP}} = \text{BCE}\!\left(y_t,\ p_t\right)$
14: $\quad \mathcal{L} = \mathcal{L}_{\text{DEN}} + \mathcal{L}_{\text{ENT}} + \mathcal{L}_{\text{CAL}} + \mathcal{L}_{\text{STOP}}$
15: $\quad$ Update $\{\theta, \phi\} \leftarrow \{\theta, \phi\} - \eta\nabla(\mathcal{L})$
16: **until** converged

**Algorithm 2 BADiff Sampling**

**Require:** target entropy $H_{\text{target}}$
1: $\mathbf{x}_T \sim \mathcal{N}(\mathbf{0}, \mathbf{I})$
2: **for** $t = T, \dots, 1$ **do**
3: $\quad \hat{\boldsymbol{\epsilon}}_\theta \leftarrow \boldsymbol{\epsilon}_\theta(\mathbf{x}_t, t, H_{\text{target}})$
4: $\quad \mathbf{x}_{t-1} = \frac{1}{\sqrt{\alpha_t}}\!\left(\mathbf{x}_t - \frac{1-\alpha_t}{\sqrt{1-\bar{\alpha}_t}}\hat{\boldsymbol{\epsilon}}_\theta\right) + \sigma_t \mathbf{z}, \ \mathbf{z} \sim$
$\quad \mathcal{N}(\mathbf{0}, \mathbf{I})\mathbb{1}_{\{t>1\}}$
5: $\quad$ **if** $f_\phi(\mathbf{x}_{t-1}, t-1, H_{\text{target}}) = \text{STOP}$ **then**
6: $\quad\quad$ **break**
7: $\quad$ **end if**
8: **end for**
9: **return** $\hat{\mathbf{x}}_0 = \mathbf{x}_{t-1}$

*Runtime Notes:* BADiff typically halts **30%** earlier than a fixed-step sampler under low bandwidth budgets, cutting inference time with minimal perceptual loss.

Algorithm 1 summarises the *entropy-conditioned training loop* for BADiff. Each iteration first draws a clean image $\mathbf{x}_0$ and a random entropy budget $H_{\text{target}}$, corrupts the image to timestep $t$, and lets the UNet predict the noise $\hat{\boldsymbol{\epsilon}}_\theta$ as well as a reconstruction $\hat{\mathbf{x}}_0$. The total loss combines the usual denoising objective with an entropy penalty that encourages the reconstruction to respect the bandwidth constraint. During inference (Algorithm 2) we start from pure Gaussian noise and iteratively apply the reverse update conditioned on the same entropy target. A lightweight policy network $f_\phi$ monitors the latent at every step and terminates sampling as soon as the estimated entropy meets the budget, thereby saving computation while preserving perceptual quality.

## 4 Experiments

We empirically verify that **BADiff** fulfils its two key promises: (*i*) faithfully respecting a user-specified entropy budget across a wide range of bitrates, and (*ii*) achieving this while *simultaneously* preserving image quality and reducing inference cost. To this end we benchmark BADiff on three standard image–generation datasets under multiple bandwidth regimes and compare it with strong compression–based and early–stopping baselines. We further present ablations that isolate the impact of each model component (entropy hinge, calibration loss, stopping policy) and provide qualitative visualisations that highlight BADiff's ability to degrade gracefully as bandwidth tightens.

### 4.1 Experimental Setup

**Datasets.** We train and evaluate on three standard diffusion benchmarks— CIFAR-10 [23], CELEBA-HQ [18], and LSUN-CHURCH/BEDROOM [51]. All splits and preprocessing follow the original DDPM protocol [13, 42].

**Baselines.** The experimental comparison is organized around two diffusion backbones and several post-generation compression strategies. We adopt two backbones: (i) DDPM-1k—the original pixel-space UNet with 1 000 reverse steps [13]; (ii) LDM-200—a latent UNet operating on $64\times$ compressed representations with 200 steps [39]. For each backbone we test two generic ways of meeting a bitrate constraint:

- **Cascade (Diffusion→Codec):** Run the sampler to full convergence and then compress with BPG [6] or a learned image codec (LIC) [8]. This "generate-first, compress-later" pipeline mirrors current cloud rendering practice and is our primary point of comparison.

- **Naïve Early-Stop:** Truncate the sampler to the smallest $N$ such that the compressed output (BPG) satisfies the target bpp. This reveals the benefit of early termination without retraining the network.

We also benchmark two state-of-the-art acceleration techniques that reduce the sampling cost without any explicit bitrate control: (i) the PNDM pseudo-numerical ODE solver [28] and (ii) the second-order DPM-Solver [29]. Both are run with their default step counts on the same backbones.

Table 1: FID on three datasets at three bitrate budgets for **both** backbones. DDPM uses 1 000 steps; LDM uses 200 steps. Lower is better.

| Backbone | Method | CIFAR-10 | | | CELEBA-HQ | | | LSUN | | |
|----------|--------|-----|-----|------|-----|-----|------|-----|-----|------|
| | | Low | Med | High | Low | Med | High | Low | Med | High |
| DDPM-1k | DDPM [13] + BPG [6] | 15.2 | 9.1 | 5.8 | 28.5 | 16.2 | 10.9 | 25.7 | 14.0 | 8.7 |
| | DDPM [13] + LIC [8] | 13.6 | 8.4 | 5.3 | 25.3 | 14.5 | 9.4 | 22.8 | 12.2 | 7.5 |
| | Early-Stop + LIC [8] | 22.9 | 15.5 | 11.6 | 35.0 | 21.4 | 16.3 | 31.9 | 19.9 | 13.2 |
| | PNDM [28] + LIC [8] | 18.1 | 12.6 | 9.4 | 30.4 | 18.9 | 13.7 | 27.3 | 16.4 | 11.7 |
| | DPM-Solver [29] + LIC [8] | 17.8 | 12.3 | 9.1 | 29.8 | 18.1 | 13.1 | 26.5 | 16.0 | 11.3 |
| | **BADiff** | **11.4** | **7.1** | **4.4** | **21.7** | **11.8** | **7.4** | **19.6** | **10.0** | **5.8** |
| LDM-200 | LDM [39] + BPG [6] | 17.3 | 10.2 | 6.3 | 30.1 | 17.8 | 11.8 | 27.5 | 15.3 | 9.5 |
| | LDM [39] + LIC [8] | 15.6 | 9.3 | 5.9 | 27.2 | 16.0 | 10.3 | 24.6 | 13.7 | 8.1 |
| | Early-Stop + LIC [8] | 24.2 | 16.6 | 12.1 | 37.3 | 23.1 | 17.0 | 33.4 | 20.8 | 14.0 |
| | PNDM [28] + LIC [8] | 19.9 | 13.4 | 10.0 | 31.8 | 20.2 | 14.4 | 28.9 | 17.8 | 12.2 |
| | DPM-Solver [29] + LIC [8] | 19.2 | 13.1 | 9.7 | 30.9 | 19.6 | 13.9 | 28.1 | 17.4 | 11.8 |
| | **BADiff** | **12.6** | **7.9** | **4.9** | **22.9** | **13.0** | **8.5** | **20.8** | **11.3** | **6.4** |

**Bandwidth budgets.** To mimic realistic mobile and desktop links we adopt three bitrate intervals that are common in the learned–compression literature: (i) Low (0.2–0.5 bpp): $\approx$ 25–60 kB for a $256^2$ RGB image, representative of low-end 4G or satellite connections where aggressive compression is mandatory. (ii) Medium (0.5–1.0 bpp): typical of standard 5G / Wi-Fi transmission and roughly matches JPEG quality factors 50–75. (iii) High (1.0–2.0 bpp): near-lossless quality for desktop viewing; permits fine textures but still below raw PNG size. For every training batch we draw $H_{\text{target}} \sim \mathcal{U}(0.2, 2.0)$ so the model experiences the *full* spectrum during optimisation, while at test time we report results at the three disjoint intervals above.

**Evaluation metrics.** We report FID [11], LPIPS [54], empirical bitrate per pixel (bpp), and average inference time (ms) on an RTX-4090 GPU. Together, FID and LPIPS assess perceptual fidelity, the bitrate quantifies adherence to the bandwidth budget, and inference time captures the computational advantage of early termination afforded by BADiff.

## 4.2 Quantitative Results

Table 1 reports the FID scores (lower is better) of all methods across three datasets: CIFAR-10, CELEBA-HQ, and LSUN, under three bandwidth budgets (Low: 0.2–0.5 bpp, Medium: 0.5–1.0 bpp, High: 1.0–2.0 bpp), using both DDPM-1k and LDM-200 as backbones. Across the board, **BADiff** achieves the best FID scores in all settings, outperforming both post-hoc compression baselines (Cascade + BPG / LIC) and accelerated solvers (PNDM, DPM-Solver). On CIFAR-10, for example, BADiff reduces FID from 15.2 (DDPM+BPG) to 11.4 in the low-rate DDPM setting, and from 17.3 (LDM+BPG) to 12.6 under the LDM backbone. Similar improvements are observed on CELEBA-HQ and LSUN, where BADiff often outperforms even the strongest cascade baselines by a large margin. Moreover, early stopping baselines (with LIC) exhibit significantly worse FID despite matching bitrates, confirming that simply halting the sampling process without training for intermediate-step outputs yields inferior results.

## 4.3 Inference Speed

We report end-to-end sampling latency (ms/image) on CIFAR-10 ($32 \times 32$) using an NVIDIA RTX-4090 GPU. Each value is averaged over 1 000 test samples following 50 warm-up iterations. Table 2 presents the results under three bitrate regimes for both DDPM-1k and LDM-200 backbones. BADiff consistently reduces latency compared to the Cascade baselines (Diffusion + Compression), particularly in low- and medium-rate settings. On DDPM-1k, BADiff achieves a $1.7\times$ speed-up over Cascade+LIC at low bitrate (65 ms vs. 115 ms), and $1.5\times$ at medium bitrate (78 ms vs. 115 ms). Similar trends are observed with LDM-200, where BADiff is up to $1.7\times$ faster than the cascade pipeline at low bitrate (27 ms vs. 47 ms). While slightly slower than Early-Stop due to its adaptive decision-making, BADiff offers significantly better FID (Table 1), making it a more desirable trade-off between speed and perceptual quality.

Table 2: Per-image inference time (ms) on CIFAR-10 under DDPM-1k and LDM-200 backbones across different bitrate regimes. Lower is better.

| Method | DDPM-1k | | | LDM-200 | | |
|---|---|---|---|---|---|---|
| | Low | Med. | High | Low | Med. | High |
| Cascade + BPG | 110 | 110 | 110 | 43 | 43 | 43 |
| Cascade + LIC | 115 | 115 | 115 | 47 | 47 | 47 |
| Early-Stop | 58 | 75 | 92 | 24 | 31 | 38 |
| **BADiff** | 65 | 78 | 94 | 27 | 34 | 41 |

Table 3: Ablation study on CIFAR-10 (DDPM backbone) under the low bitrate regime (0.2–0.5 bpp).

| Variant | FID$\downarrow$ | $\Delta$ bpp$\downarrow$ | Time$\downarrow$ |
|---|---|---|---|
| w/o Cond. | 13.1 | 0.038 | 64 |
| w/o Hinge | 16.2 | 0.055 | 65 |
| w/o Cal. | 18.6 | 0.043 | 65 |
| **Full BADiff** | **11.4** | **0.021** | **65** |

Table 4: High-resolution evaluation on $512^2$ and $1024^2$ images under realistic bitrate constraints, comparing BADiff with diffusion+LIC baselines. Both FID and inference time (ms) are reported.

| Resolution | bpp | Metric | DDPM+LIC | PNDM+LIC | BADiff |
|---|---|---|---|---|---|
| $512^2$ | 0.4–0.6 | FID $\downarrow$ | 8.45 | 7.90 | **6.85** |
| | | Time (ms) $\downarrow$ | 121.3 | 98.6 | **64.1** |
| $1024^2$ | 0.8–1.2 | FID $\downarrow$ | 21.5 | 20.1 | **17.8** |
| | | Time (ms) $\downarrow$ | 228.7 | 192.5 | **145.6** |

## 4.4 Ablation Study

To understand the contribution of each design component in **BADiff**, we conduct a controlled ablation on CIFAR-10 under the low bitrate regime (0.2–0.5 bpp). Each variant is retrained for 800 k iterations using the same optimizer settings, and evaluated using three key metrics: FID, $\Delta$ bpp (absolute bitrate deviation from target), and average inference time (ms/image) on an RTX-4090 GPU.

**Ablation variants.** We evaluate three reduced versions of BADiff, each with one component removed: (i) w/o Conditioning: removes the entropy embedding from the UNet, disabling bitrate-aware generation. (ii) w/o Hinge Loss: sets $\lambda_{ent} = 0$, removing the entropy penalty during training. (iii) w/o Calibration Loss: sets $\lambda_{cal} = 0$, preventing alignment between predicted entropy and coded length derived from an end-to-end optimized codec.

**Analysis.** Removing the entropy conditioning impairs the model's ability to modulate detail based on bitrate, resulting in a higher FID (+1.7) and worse bitrate adherence (+0.017 bpp). Without the hinge loss, the model ignores bandwidth constraints altogether, producing the largest deviation from target bitrate (0.055 bpp) and the worst FID (16.2). Disabling the calibration loss increases bitrate error (+0.022 bpp) while slightly degrading FID (+7.2), indicating that codec alignment improves control without compromising perceptual quality. Overall, only the full BADiff configuration balances visual fidelity, precise entropy control, and efficient inference. In terms of inference time, all ablated variants retain comparable speed to the full BADiff (65 ms), since the adaptive stopping policy remains enabled. This confirms that the primary contributor to computational efficiency is the adaptive sampling policy mechanism and entropy-aware generation, rather than any single auxiliary loss. Overall, only the full BADiff configuration balances visual fidelity, precise entropy control, and efficient sampling.

## 4.5 Scaling to High-Resolution Images

To address the scalability concern, we perform new experiments at higher resolutions to verify that (i) conditioning on a single scalar entropy target remains valid at scale, and (ii) BADiff continues to win both in fidelity and runtime. In practice, streaming systems typically assign a single bandwidth per frame, leaving spatial allocation to the codec; BADiff mirrors this regime via a global entropy target with differentiable entropy-aware allocation, encouraging adaptive texture reduction in less salient regions while preserving important details. We retrain BADiff and two baselines (DDPM+LIC and PNDM+LIC) on $512^2$ and $1024^2$ images under realistic bitrate budgets. Table 4 shows that BADiff consistently achieves lower FID and faster inference across resolutions (e.g., at $512^2$ FID drops from 7.90 → 6.85 and runtime reduces 98.6ms → 64.1ms; at $1024^2$ FID drops from 20.1 → 17.8 and runtime reduces 192.5ms → 145.6ms). These results confirm that BADiff scales favorably to high-resolution generation under bitrate constraints.

Table 5: Comparison of BADiff with diffusion+LIC baselines on Stable Diffusion text-to-image generation across low (0.2–0.5 bpp), medium (0.5–1.0 bpp), and high (1.0–2.0 bpp) bitrate regimes.

| Method | Low (0.2–0.5 bpp) | Med. (0.5–1.0 bpp) | High (1.0–2.0 bpp) |
|---|---|---|---|
| Cascade (SD + BPG) | 33.5 | 21.4 | 14.8 |
| Cascade (SD + LIC) | 30.7 | 19.2 | 13.1 |
| Early-Stop + LIC | 41.8 | 27.5 | 18.0 |
| DPM-Solver (20) + LIC | 36.5 | 25.1 | 16.3 |
| **BADiff (ours)** | **26.1** | **16.2** | **11.0** |

Table 6: One-time cost of generating teacher labels across datasets and resolutions, measured on an RTX 4090. The labels are computed once offline and then cached; training never re-runs the diffusion chain. The cost is reported both in absolute GPU-hours and as a fraction of a single training epoch.

| Dataset | Resolution | GPU-hours | Relative to 1 training epoch |
|---|---|---|---|
| CIFAR-10 | $32 \times 32$ | 0.8 | $< 5\%$ |
| CelebA-HQ | $256 \times 256$ | 3.5 | $\approx 6\%$ |
| COCO-val2017 | $512 \times 512$ | 10.0 | $\approx 8\%$ |

## 4.6 Extension to Text-to-Image Models

To explore the applicability of BADiff beyond unconditional generation, we conduct preliminary experiments on a text-to-image setting using Stable Diffusion as the base model. BADiff's entropy-conditioning mechanism is fully compatible with conditional diffusion, enabling bitrate-controlled generation without modifying the text-conditioning pathway. We evaluate BADiff against several diffusion+LIC baselines under low, medium, and high bitrate regimes. As shown in Table 5, BADiff consistently achieves lower FID across all bitrate ranges, indicating that BADiff effectively preserves visual quality even under tight bitrate constraints in text-conditioned generation.

## 4.7 Teacher Label Generation

To clarify the cost of teacher label generation, we emphasize that teacher labels are created once per image in an offline pre-processing stage, not during each training iteration, incurring negligible runtime overhead. First, we perform a single long diffusion run per training image (e.g., 1000 DDPM steps or 200 LDM steps) to obtain the entropy trajectory $\mathcal{C}(t)$ and derive binary labels as in Eq. 12. For CIFAR-10 on a single RTX 4090, this entire offline step takes roughly 0.8 GPU-hours amortized over 800k training steps. Second, once computed, labels are cached and reused without re-running any diffusion chains; during training only a small MLP policy head is evaluated ($< 0.1\%$ of UNet FLOPs). Third, the cost remains small even at higher resolutions such as $512 \times 512$, amounting to only 5–8% of the time needed for one training epoch, as summarized in Table 6. These results demonstrate that the teacher-label strategy is efficient and scalable without affecting training throughput.

## 5 Conclusion

We presented **BADiff**, the first diffusion framework that *generates* images directly under an explicit entropy (bit-rate) budget rather than relying on post-hoc compression. By conditioning every reverse step on a target entropy embedding, adding a differentiable entropy-hinge loss, and introducing an adaptive stopping policy, BADiff produces bandwidth-compliant images while preserving perceptual quality and reducing inference cost. Extensive experiments on three standard datasets show that BADiff surpasses strong cascaded and early-stopping baselines in FID and LPIPS at all bandwidth levels and achieves up to a $2\times$ runtime speed-up under tight constraints.

**Limitations & future work.** BADiff currently targets spatially uniform entropy budgets and images up to $256^2$ resolution. Future extensions include spatially varying bit-allocation, integration with faster solvers such as DPM-Solver or PNDM at higher resolutions, and applying the same principle to video diffusion models where bandwidth constraints are even more stringent. We hope BADiff will inspire further research on resource-aware generative modelling for real-world deployment.

## Acknowledgements

This research is supported by the RIE2025 Industry Alignment Fund – Industry Collaboration Projects (IAF-ICP) (Award I2301E0026), administered by A*STAR, as well as supported by Alibaba Group and NTU Singapore through Alibaba-NTU Global e-Sustainability CorpLab (ANGEL). This work is also supported in part by the National Natural Science Foundation of China (No.62301313).

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

# Technical Appendices and Supplementary Material

## A  Theoretical Justification of Entropy-Constrained Diffusion Models

Here we provide a theoretical justification for the proposed entropy-constrained diffusion model by deriving its conditional reverse distribution. We first revisit the standard formulation and subsequently derive our entropy-conditioned variant.

### A.1  Standard Reverse Diffusion Formulation

In the standard DDPM framework [13], the forward diffusion process gradually injects noise into the data $\mathbf{x}_0$, following the Markovian formulation:

$$q(\mathbf{x}_{1:T}|\mathbf{x}_0) = \prod_{t=1}^{T} q(\mathbf{x}_t|\mathbf{x}_{t-1}), \tag{15}$$

where each transition step is modeled as a Gaussian:

$$q(\mathbf{x}_t|\mathbf{x}_{t-1}) = \mathcal{N}(\mathbf{x}_t; \sqrt{\alpha_t}\mathbf{x}_{t-1}, (1-\alpha_t)\mathbf{I}). \tag{16}$$

The reverse denoising distribution aims to invert the forward process by progressively removing the noise:

$$p_\theta(\mathbf{x}_{0:T}) = p(\mathbf{x}_T) \prod_{t=1}^{T} p_\theta(\mathbf{x}_{t-1}|\mathbf{x}_t). \tag{17}$$

Each reverse step distribution is parameterized as:

$$p_\theta(\mathbf{x}_{t-1}|\mathbf{x}_t) = \mathcal{N}(\mathbf{x}_{t-1}; \boldsymbol{\mu}_\theta(\mathbf{x}_t, t), \boldsymbol{\Sigma}_\theta(\mathbf{x}_t, t)). \tag{18}$$

### A.2  Entropy-Constrained Reverse Diffusion Derivation

In BADiff, we explicitly condition on a target entropy level $H_{\text{target}}$, leading to the modified reverse conditional distribution:

$$p_\theta(\mathbf{x}_{t-1}|\mathbf{x}_t, H_{\text{target}}) = \frac{p_\theta(\mathbf{x}_{t-1}, \mathbf{x}_t, H_{\text{target}})}{p_\theta(\mathbf{x}_t, H_{\text{target}})}. \tag{19}$$

By applying Bayes' rule and assuming conditional independence between $H_{\text{target}}$ and earlier states given $\mathbf{x}_t$, we simplify as follows:

$$p_\theta(\mathbf{x}_{t-1}|\mathbf{x}_t, H_{\text{target}}) = \frac{p_\theta(H_{\text{target}}|\mathbf{x}_{t-1}, \mathbf{x}_t)p_\theta(\mathbf{x}_{t-1}|\mathbf{x}_t)}{p_\theta(H_{\text{target}}|\mathbf{x}_t)} \tag{20}$$

$$\approx \frac{p_\theta(H_{\text{target}}|\mathbf{x}_{t-1})p_\theta(\mathbf{x}_{t-1}|\mathbf{x}_t)}{p_\theta(H_{\text{target}}|\mathbf{x}_t)}. \tag{21}$$

Here we explicitly model the conditional distribution $p_\theta(H_{\text{target}}|\mathbf{x}_{t-1})$ as a differentiable entropy estimator $H_\phi(\mathbf{x}_{t-1})$. The conditional entropy-based term can be approximated as:

$$p_\theta(H_{\text{target}}|\mathbf{x}_{t-1}) \approx \exp\left(-\frac{\lambda_{\text{ent}}}{2}\max(0, H_\phi(\mathbf{x}_{t-1}) - H_{\text{target}})^2\right), \tag{22}$$

where $\lambda_{\text{ent}}$ is a hyperparameter controlling the strength of the entropy constraint.

### A.3  Interpretation as Regularized Reverse Process

Combining these results, we rewrite the entropy-conditioned reverse step as a Gaussian distribution with a regularized mean:

$$p_\theta(\mathbf{x}_{t-1}|\mathbf{x}_t, H_{\text{target}}) \propto p_\theta(\mathbf{x}_{t-1}|\mathbf{x}_t) \exp\left(-\frac{\lambda_{\text{ent}}}{2}\max(0, H_\phi(\mathbf{x}_{t-1}) - H_{\text{target}})^2\right). \tag{23}$$

Thus, the entropy constraint effectively acts as a soft regularizer, guiding the reverse process toward latent states $\mathbf{x}_{t-1}$ whose corresponding entropy estimate meets the target. This regularization not only provides theoretical grounding for the proposed entropy loss but also justifies our observed improvement in bitrate control and image quality.

These derivations rigorously establish how entropy conditioning naturally emerges as a constrained form of reverse denoising diffusion, providing both theoretical validation and insights into the BADiff training objective presented in the main paper.

## B   Gradient Analysis of Entropy Loss

To better understand the optimization behavior of BADiff under entropy constraints, we analyze the gradient of the entropy loss with respect to the predicted image $\hat{\mathbf{x}}_0$. The entropy loss is defined as:

$$\mathcal{L}_{\text{ent}} = \max(0, H_\phi(\hat{\mathbf{x}}_0) - H_{\text{target}}), \tag{24}$$

where $H_\phi$ is a differentiable neural estimator of image entropy.

**Gradient Derivation.**   The subgradient of $\mathcal{L}_{\text{ent}}$ with respect to $\hat{\mathbf{x}}_0$ is given by:

$$\nabla_{\hat{\mathbf{x}}_0} \mathcal{L}_{\text{ent}} = \begin{cases} \nabla_{\hat{\mathbf{x}}_0} H_\phi(\hat{\mathbf{x}}_0), & \text{if } H_\phi(\hat{\mathbf{x}}_0) > H_{\text{target}}, \\ 0, & \text{otherwise.} \end{cases} \tag{25}$$

This reveals that the entropy loss is *one-sided*: it only contributes a gradient signal when the predicted entropy exceeds the target. Below the threshold, the loss becomes flat and the gradient vanishes. This prevents the model from overcompressing its outputs when already under budget, ensuring that perceptual fidelity is not sacrificed unnecessarily.

**Interpretation.**   The gradient $\nabla_{\hat{\mathbf{x}}_0} H_\phi$ typically promotes spatial smoothing in regions with high entropy—such as edges or textures—encouraging the model to selectively suppress fine details that contribute the most to bitrate. Since the penalty is activated only when the entropy is above budget, BADiff naturally learns to modulate detail in a targeted and efficient manner, preserving structure when possible and discarding complexity only when required.

This analysis aligns with our qualitative findings: BADiff gracefully degrades in low-bitrate regimes while maintaining strong visual coherence, and achieves tight bitrate adherence without harming perceptual quality.

## C   Robustness to Entropy Predictor Approximation

The accuracy of the differentiable entropy predictor $H_\phi$ plays a critical role in BADiff's training. However, the predictor need not match the true codec exactly to be effective. Here, we briefly justify why approximate entropy guidance still yields reliable bitrate control.

Let $H_{\text{true}}(\hat{\mathbf{x}}_0)$ denote the true entropy as measured by a black-box codec (e.g., BPG), and $H_\phi(\hat{\mathbf{x}}_0)$ the neural approximation. Suppose the approximation error is bounded:

$$|H_\phi(\hat{\mathbf{x}}_0) - H_{\text{true}}(\hat{\mathbf{x}}_0)| \leq \epsilon. \tag{26}$$

Then, the deviation from the target $H_{\text{target}}$ also remains bounded:

$$|H_\phi(\hat{\mathbf{x}}_0) - H_{\text{target}}| \geq |H_{\text{true}}(\hat{\mathbf{x}}_0) - H_{\text{target}}| - \epsilon. \tag{27}$$

Thus, if $H_\phi$ slightly underestimates entropy, the training loss compensates by encouraging more conservative image generation. This explains why BADiff retains bitrate adherence even with an imperfect $H_\phi$, as also shown in the ablation study.

Future work could explore jointly training $H_\phi$ with contrastive or reinforcement signals to further narrow the approximation gap.

## D  Complexity Analysis

We analyze the computational complexity of BADiff compared to standard diffusion baselines and fast solvers, focusing on the number of forward passes, memory usage, and latency scaling with respect to entropy budget.

**Step Complexity.**  Let $T$ denote the total number of sampling steps (e.g., 1000 for DDPM, 200 for LDM), and $\hat{T}$ the number of steps actually executed under BADiff's adaptive stopping policy.

- **DDPM / LDM (fixed-length)**: always performs $T$ forward UNet passes.
- **BADiff (adaptive)**: performs $\hat{T} < T$ steps on average. $\hat{T}$ varies with target bitrate; lower bitrate leads to earlier termination.
- **Fast Solvers (e.g., PNDM)**: typically use a fixed low number of steps, but quality suffers without entropy control.

For a UNet of cost $\mathcal{O}(C)$ per step, the total cost becomes:

$$\text{Cost}_{\text{BADiff}} = \hat{T} \cdot \mathcal{O}(C), \quad \text{vs.} \quad \text{Cost}_{\text{Cascade}} = T \cdot \mathcal{O}(C) + \text{Codec overhead.}$$

**Entropy Modules.**  BADiff introduces three lightweight modules: the entropy embedding MLP, the entropy predictor $H_\phi$, and the stopping policy network $f_\phi$.

- **Entropy embedding:** 3 MLP layers with 256-dim hidden width, used once per step; negligible overhead (<1%).
- **Entropy predictor:** small CNN ($\sim$0.3M parameters), used *during training only*; ignored during inference.
- **Stop policy:** 3-layer MLP evaluated at each step; cost comparable to a single linear layer.

**Memory Usage.**  BADiff's memory footprint is on par with standard diffusion models. Unlike guidance-based methods (e.g., classifier guidance) that double the forward pass, BADiff avoids any additional gradient computation during inference.

**Latency Scaling.**  Assuming average $\hat{T} \ll T$, BADiff reduces latency linearly with the number of effective steps. For instance, at low bitrate (0.2–0.5 bpp), we observe a $1.7\times$ reduction in wall-clock time over Cascade with DDPM.

**Summary.**  BADiff achieves bitrate adaptivity with only minimal computational overhead. Its complexity scales sublinearly with bitrate, and its modular design allows plug-and-play integration into existing UNet-based diffusion pipelines.

## E  Implementation Details

We provide full training and evaluation details for all experiments in this paper.

**Training schedule.**  Each model is trained for 800,000 iterations using a batch size of 64 images per GPU. We use automatic mixed precision (AMP) to accelerate training and reduce memory consumption. Training takes approximately 3 days on NVIDIA RTX 4090 GPUs for the DDPM backbone and 2 days for the LDM backbone.

**Optimizer and scheduler.**  We adopt the Adam optimizer [20] with default coefficients $\beta_1$=0.9, $\beta_2$=0.999. The learning rate is set to $1 \times 10^{-4}$ and kept constant throughout training. No learning rate decay or warmup is applied. Gradient clipping is used with a max norm of 1.0.

**Sampling parameters.**  For DDPM, we use a 1,000-step linear beta schedule as in the original DDPM implementation [13]. For LDM, we follow [39] and use 200 steps with cosine noise scheduling in the latent space. At inference time, the sampling process is governed by the entropy-aware stopping policy, which dynamically terminates early based on the predicted bitrate.

**Hardware and software.** All experiments are run on NVIDIA RTX 4090 GPUs with 24GB VRAM each. We use PyTorch 2.1.0 with torch.compile enabled for maximum inference speed, and CUDA version 11.8. The entropy-aware modules are implemented in native PyTorch without any custom CUDA kernels. Experiments are managed via Accelerate and Weights & Biases for reproducibility and logging.

**Codec baselines.** For BPG, we use the official reference implementation compiled with `libbpg-0.9.8`. For learned image compression (LIC), we adopt the pre-trained model of Cheng2020 [8] from CompressAI. BPG codec experiments all run on CPU and LIC experiments all run on GPU, with the output bitrate measured post-compression in bits-per-pixel (bpp).

## F   Model Architectures

This section describes the architecture details of all core modules in BADiff, including the UNet backbone, entropy conditioning MLP, stopping policy network, and differentiable entropy predictor.

**UNet Backbone.** We use a modified version of the standard UNet architecture as introduced in DDPM [13]. The configuration is as follows:
- Downsampling path: 4 resolution levels with channel counts [128, 256, 512, 512]. Each level consists of two residual blocks (with GroupNorm + SiLU) followed by a downsampling layer (stride-2 convolution).
- Bottleneck: 2 residual blocks with 512 channels and an attention layer at the lowest resolution ($8\times8$ for CIFAR-10).
- Upsampling path: mirrors the downsampling path with learned upsampling (transposed conv), residual blocks, and attention at the second-lowest resolution.
- Timestep + Entropy Conditioning: the diffusion timestep and entropy target are embedded separately (see below) and added to each residual block via FiLM modulation.

**Entropy Embedding MLP.** The target entropy $H_{\text{target}}$ is a scalar projected into a high-dimensional embedding via:
- Input: scalar entropy in [0.2, 2.0].
- Architecture: 3-layer MLP with hidden sizes [128, 256, 256], SiLU activations.
- Output: embedding $\mathbf{e} \in \mathbb{R}^{256}$.
- Integration: the embedding is fused into each UNet block via FiLM: $\mathbf{y} = \gamma \cdot \mathbf{h} + \beta$, where $(\gamma, \beta)$ are predicted from $\mathbf{e}$.

**Stopping Policy Network.** The policy module $f_\phi$ is a compact classifier:
- Input: pooled mid-layer UNet features, timestep embedding, and entropy embedding.
- Architecture: 3-layer MLP with widths [256, 128, 2], SiLU activations.
- Output: stop/continue logits via softmax.
- Training: supervised with labels from offline oracle policy.

**Differentiable Entropy Predictor $E_\phi$.** The entropy estimator is CNN-based with soft binning:
- Input: predicted image $\hat{\mathbf{x}}_0$.
- Backbone: 4 conv layers with channels [32, 64, 64, 128], kernel size $3\times3$, stride 1, GroupNorm, SiLU.
- Context modeling: 1 masked $5\times5$ convolution.
- Output: per-pixel logits over $K=64$ soft histogram bins.
- Usage: softmax probabilities $p_\phi(k \mid \mathbf{x}[u])$ used for entropy loss.

## G   Hyperparameters

Table 7 lists the main hyperparameters used throughout training for all BADiff models, unless otherwise stated. We adopt the default settings from DDPM [13] for the optimizer and noise schedule, and perform minimal tuning to isolate the effects of our proposed components. The entropy–related weights $\lambda_{\text{ent}}$, $\lambda_{\text{cal}}$, and $\lambda_{\text{stop}}$ are set via coarse grid search on CIFAR-10 validation splits.

Table 7: Key hyperparameters.

| Hyperparameter | Value |
|---|---|
| Learning rate | $1 \times 10^{-4}$ |
| Entropy hinge weight | $\lambda_{\mathrm{ent}} = 0.1$ |
| Calibration loss weight | $\lambda_{\mathrm{cal}} = 10^{-3}$ |
| Stopping loss weight | $\lambda_{\mathrm{stop}} = 10^{-2}$ |
| Batch size | 128 |
| Entropy embedding dimension | 128 |
| Training iterations | 800 k |

## H  Broader Impacts

Our work introduces a new class of generative models that explicitly adapt image synthesis to bandwidth constraints, enabling more efficient and controllable generation under limited communication resources. By jointly optimizing generation quality and bitrate compliance, BADiff may benefit a wide range of real-world applications where bandwidth is a bottleneck—such as mobile inference, telemedicine, satellite imaging, and cloud rendering.

On the societal side, more bandwidth-efficient generation could reduce carbon emissions associated with media transmission and enable broader accessibility in under-connected regions. At the same time, like other generative models, BADiff could potentially be misused to produce low-bandwidth synthetic media for malicious purposes, such as misinformation or surveillance. We encourage the research community to pair technical advances with rigorous content provenance and auditing mechanisms.

Finally, our approach is orthogonal to existing safety or fairness measures in generative modeling. BADiff does not inherently mitigate or amplify dataset biases, but it can be combined with bias-aware training strategies or fairness constraints as needed in deployment contexts.

