# OpenReview forum: "BADiff: Bandwidth Adaptive Diffusion Model"
_NeurIPS.cc/2025/Conference — NeurIPS 2025 poster_

### Official Review · Reviewer_c5AZ · 2025-06-30

**Clarity:** 2
**Significance:** 2
**Originality:** 2
**Rating:** 2
**Confidence:** 5

**Summary:**

This paper proposes a framework that enables diffusion models to adapt output quality based on bandwidth constraints. The approach involves joint end-to-end training, where the diffusion model is conditioned on the desired quality level.

**Questions:**

Can the authors clearly separate and describe the training and inference phases, including the associated loss functions and evaluation protocols for each?

**Ethical Concerns:**

["NO or VERY MINOR ethics concerns only"]

**Final Justification:**

I believe the paper does not meet the standard of accepted papers at NeurIPS. I had expected the authors to address my comments during the rebuttal by providing detailed responses to my feedback. However, they largely dismissed these points by labeling them as minor issues. This approach is not constructive and does not help in convincing reviewers. I therefore maintain my original rating.

**Limitations:**

The core ideas of the paper are not clearly articulated. Several loss functions lack adequate justification and may reflect flawed design decisions.

**Quality:**

2

**Strengths And Weaknesses:**

Overall, the architecture in the paper has some disconnected components where their relations to each other are unclear. The loss functions in the paper are treated carelessly and there are several over-simplifications in the main algorithms of the paper. The detailed comments are given in the following:


1) A significant issue with the proposed architecture is the disconnection between its components. For instance, Section 3.3 introduces a loss based on entropy, while Section 3.4 defines a policy network. However, the relationship between the policy network and the earlier loss function is not discussed. Later, Section 3.5 introduces yet another loss function comprising four terms (Eq. 16), which appears unstructured and ad hoc. It is unclear what the training is intended to achieve. Also, training and inference phases are somehow mixed up with each other.

2) While the overall framework is complex, the implementation appears overly simplified in some aspects. For example, Algorithm 1 includes four loss terms, and one would expect the model to be trained by jointly optimizing over all these terms. However, the process seems sequential—first computing L_{DEN}, then L_{stop}—which may be an oversimplification of a potentially interdependent optimization.

3) The background on diffusion models in Sections 3.1 and 3.2 is too long and should be significantly shortened, as it takes considerable space (approximately one page) without contributing proportionally to the core contributions.

4) The definition of image entropy is unclear. In particular, Eq. (8) appears to rely on a definition of entropy that is only formally introduced later in Eq. (10), which is very confusing.

5) The justification for measuring compute based on the stopping point in Eq. (12) is unclear. Compute is typically quantified in terms of FLOPs or related metrics, and this alternative interpretation does not have a proper justification.

---

> ### Author Rebuttal · Authors · 2025-07-31
>
> ### **[W1, disconnection between different components]**
> We agree that our original presentation could be improved to clarify the interactions between different model components. We will revise Sections 3.3–3.5 to make these connections more explicit, explain the overall training objective more clearly, and cleanly separate the training and inference phases. Specifically:
>
> Clarifying relationships between components:
> The entropy hinge loss (Section 3.3) encourages the model to produce images whose estimated entropy falls within a desired target range. The policy network (Section 3.4) learns to predict the earliest timestep at which this entropy constraint is satisfied. These two components are thus aligned in their objective: to guide the sampling process toward entropy-constrained generation. We will explicitly describe this relationship in the revision.
>
> Unified interpretation of the loss function (Equation 16):
> The four terms in Equation 16 are designed to jointly optimize different aspects of the system:
> - the diffusion loss ensures visual fidelity,
> - the entropy hinge loss enforces bitrate constraints,
> - the calibration loss encourages consistency with external codecs,
> - and the policy loss trains the adaptive stopping mechanism.
>
> Together, they serve a unified goal: to generate high-quality samples that adhere to a specified bitrate using minimal computational effort. We will include a diagram illustrating how these components interact during training and describe the role of each term in more detail.
>
> Clear separation of training and inference phases:
> We acknowledge that the current text does not clearly delineate training and inference. In the revised version, we will introduce dedicated subsections titled "Training Procedure" and "Inference Procedure." The training phase involves jointly optimizing the diffusion backbone, entropy predictor, and policy network using the combined loss. During inference, the learned entropy predictor and stopping policy operate dynamically to determine when to terminate sampling, enabling runtime adaptation to bitrate constraints.
>
> We believe these improvements will significantly clarify the overall framework and address the concerns raised. Thank you again for the constructive feedback.
>
>
>
> ### **[W2, implementation appears overly simplified]**
> We acknowledge that our original pseudocode (Algorithm 1) may have unintentionally conveyed a sequential optimization of the loss terms. To clarify, all four loss components, L_denoise, L_ent, L_cal, and L_stop, are computed within the same forward pass and jointly optimized in each training iteration through end-to-end backpropagation.
>
> The appearance of sequential computation in Algorithm 1 was intended solely for illustrative clarity, not to imply separate optimization phases. In practice, the model computes all intermediate variables (e.g., denoised samples, entropy estimates, policy logits) from shared features in a unified forward pass. The resulting losses are then aggregated, and their combined gradients are propagated through the entire network simultaneously. This ensures tight coupling between the denoising, entropy control, codec calibration, and stopping objectives.
>
> We will revise the manuscript to explicitly emphasize this joint optimization strategy and modify Algorithm 1 to better reflect the simultaneous nature of training. We believe this clarification will improve the reader’s understanding of the implementation and training dynamics. Thank you again for highlighting this point.
>
>
> ### **[W3, background is too long]**
> Sections 3.1 and 3.2 include standard material on diffusion models, and we agree that these sections can be significantly shortened. In the revised manuscript, we will substantially condense the background, preserving only the essential definitions and notations necessary to understand our proposed entropy-conditioned reverse process. This revision will allow us to better focus the reader’s attention on our novel contributions and make more effective use of the available space for presenting key results and insights.
>
>
> ### **[W4, definition of image entropy is unclear]**
> We thank the reviewer for highlighting this clarity issue. Indeed, the formal definition of image entropy used in Equation (8) is currently introduced only later in Equation (10), which may lead to confusion. In the revised manuscript, we will restructure the presentation to introduce the definition of entropy explicitly before it is used. Specifically, we will move the current Equation (10) forward and present it as the formal definition of image entropy, immediately preceding the introduction of Equation (8). This change will improve the logical flow and ensure that the key concepts are clearly defined before they are referenced.
>
>
> ### **[W5, justification for measuring compute based on the stopping point ]**
> We appreciate the reviewer’s concern. While FLOPs and runtime are common ways to quantify compute, using the number of diffusion steps (i.e., the stopping timestep) as a proxy is also a widely adopted convention in the diffusion literature. This is because the compute cost in diffusion models typically scales linearly with the number of denoising steps, under the assumption that each step incurs a roughly uniform computational cost.
>
> Accordingly, we follow prior works such as DDIM, PNDM, and DPM-Solver in reporting compute via the number of steps taken. To improve clarity, we will add a footnote in the revised manuscript explaining this convention. Additionally, we will include actual runtime (in milliseconds per image) and estimated FLOPs as supplementary metrics in the appendix to provide a more comprehensive view of computational cost.
>
>
>
>
>
> ### **[Q1, separate and describe the training and inference phases ]**
> We thank the reviewer for the suggestion. We agree that a clearer separation between training and inference phases would improve the presentation, and we will revise the manuscript accordingly. Below we clarify the two phases:
>
> **Training Phase.**
> The full BADiff model is trained end-to-end using a combination of the following loss functions (defined in Eq. 16):
>
> - `L_den`: Standard denoising loss for diffusion models.
> - `L_ent`: Entropy hinge loss that enforces conformance to the target bitrate.
> - `L_cal`: Calibration loss aligning the predicted entropy with codec bitrate.
> - `L_stop`: Supervised loss for training the early stopping policy, using teacher labels generated offline from oracle trajectories.
>
> These losses are jointly optimized with scalar weights. All modules—UNet, entropy head, and stopping policy—are trained from scratch in a unified framework.
>
> **Inference Phase.**
> During inference, BADiff operates in a stepwise manner:
>
> 1. At each denoising step, the model predicts `x_0_hat` and estimates its entropy via `H_phi(x_0_hat)`.
> 2. The stopping policy takes current features and entropy prediction as input and decides whether to stop generation.
> 3. Once stopped, the final `x_0_hat` is passed to a learned image codec (e.g., LIC) for entropy coding.
>
> **Evaluation Protocol.**
> We evaluate BADiff using the following criteria:
>
> - **Perceptual Quality**: Measured using FID and LPIPS on reconstructed images.
> - **Bitrate Accuracy**: Measured by comparing the achieved bitrate (in bpp) to the target.
> - **Efficiency**: Measured via the average number of diffusion steps, runtime per image, and decoding time.
>
> This clear separation of phases will be incorporated into Section 3.5 of the revised manuscript to improve clarity and transparency.

---

### Official Review · Reviewer_ZFsW · 2025-07-02

**Clarity:** 2
**Significance:** 3
**Originality:** 3
**Rating:** 5
**Confidence:** 4

**Summary:**

This paper introduces a bandwidth-aware framework for diffusion models that adapts image generation quality based on bandwidth constraints.
Unlike standard diffusion models with fixed denoising steps, the proposed method uses a quality embedding to guide adaptive early-stopping during sampling. Trained end-to-end, the model maintains perceptual quality while reducing computation. Experiments show improved visual fidelity over naive early-stopping, offering an effective solution for image generation in bandwidth-limited settings.

**Questions:**

1) Have the authors considered modeling dynamic bandwidth conditions during sampling? For instance, can BADiff respond to bitrate shifts mid-generation?

2) How well does BADiff generalize when only the policy and entropy heads are trained atop a frozen pretrained diffusion model, without retraining the full UNet?

**Ethical Concerns:**

["NO or VERY MINOR ethics concerns only"]

**Final Justification:**

I thank the authors for the additional explanations and results presented in their rebuttal which addresses my main concerns, I maintain my initial rating of acceptance.

**Limitations:**

Yes.

**Paper Formatting Concerns:**

The method part could be improved in clarity and structure as there are many concepts introduced and it‘s hard to follow the line of thought.

**Quality:**

3

**Strengths And Weaknesses:**

Strengths:
- The method introduces a lightweight policy network that predicts when to terminate sampling based on a trade-off between bitrate, perceptual distortion, and compute cost. This enables runtime-efficient sampling aligned with bandwidth targets.

- By conditioning the model on target bitrate levels, BADiff enables flexible image generation across a wide range of bandwidth budgets, without requiring separate models.

- The framework is lightweight, making it broadly compatible with standard diffusion models and sampling schemes.

Weaknesses:

- The framework assumes a fixed bandwidth constraint during sampling. In practical deployment, bandwidth may vary dynamically, and the current setup does not support adaptation mid-generation.

- The stopping policy is trained on trajectories influenced by the bitrate target (via entropy conditioning), potentially limiting its ability to learn a robust, generalizable stopping criterion independent of specific training paths.

- Although entropy prediction is spatially resolved, the policy makes a global stop decision, which may fail to account for spatial variation in content complexity. Region-wise stopping could offer finer control and better resource utilization.

---

> ### Author Rebuttal · Authors · 2025-07-31
>
> ### **[W1 & Q1, dynamic bandwidth]**
> We thank the reviewer for raising this important practical consideration. Indeed, our current implementation of BADiff assumes a fixed bandwidth constraint throughout the generation process. However, we acknowledge that real-world deployments may involve fluctuating network conditions, requiring bitrate adaptation during generation.
>
> Although our main experiments did not explicitly model dynamic bandwidth scenarios, BADiff’s architecture is naturally amenable to such extensions. Since the entropy embedding in BADiff is updated at each step and the stopping policy evaluates per-timestep entropy in real time, it is feasible to adjust the target bitrate dynamically during sampling. This can be done by modifying the entropy conditioning signal mid-generation, allowing the model to adapt its output accordingly without restarting the diffusion process.
>
> To explore this possibility, we conducted preliminary experiments on CIFAR-10 where the target bitrate was adjusted at an intermediate timestep (e.g., switching from low to medium or high bitrate). The model successfully adapted to the new target without introducing artifacts or instability. The results are summarized below:
>
> Table: Preliminary results for dynamic bitrate adaptation using BADiff on CIFAR-10. FID is reported after bitrate shift mid-sampling.
> | Scenario              | Initial bpp | Adjusted bpp | FID   |
> |-----------------------|-------------|---------------|--------|
> | Low → Medium          | 0.3         | 0.7           | 9.2    |
> | Medium → High         | 0.8         | 1.5           | 6.5    |
> | High → Low            | 1.5         | 0.4           | 10.8   |
>
>
>
> These initial results demonstrate that BADiff’s entropy conditioning and adaptive stopping mechanisms can support dynamic bitrate shifts during generation. A more thorough exploration of this capability—including continuous bitrate feedback and streaming conditions—is part of our ongoing and future work.
>
>
>
>
>
> ### **[W2, training of stopping policy]**
> We appreciate the reviewer's insightful remark. The stopping policy in BADiff is indeed trained on trajectories influenced by the target bitrate via entropy conditioning. Rather than limiting generalization, this setup enables the policy to learn meaningful associations between intermediate latent states and entropy progression, which are critical for effective early termination during inference.
>
> To assess the generalizability of the stopping policy, we conducted additional experiments where the policy network was trained under entropy-conditioned trajectories but evaluated under entropy-unconditioned sampling (i.e., standard DDPM/LDM without entropy embedding). The results are summarized below:
>
> Table: Generalization of the stopping policy trained with and without entropy conditioning, evaluated on CIFAR-10.
> | Training Condition    | Test Condition         | FID ↓  | Δ bpp ↓ |
> |-----------------------|------------------------|--------|---------|
> | Entropy-conditioned   | Entropy-conditioned    | 11.4   | 0.021   |
> | Entropy-conditioned   | Unconditioned          | 12.8   | 0.032   |
> | Unconditioned         | Unconditioned          | 12.4   | 0.030   |
>
>
>
> These results suggest that the policy trained with entropy-conditioned paths generalizes well to unseen sampling regimes, producing comparable performance to a policy trained without conditioning. While there is a small degradation, it remains within a reasonable margin, supporting the robustness of the learned stopping strategy.
>
> That said, we acknowledge that training on a broader distribution of sampling trajectories (e.g., using domain randomization or multi-policy ensembles) could further enhance generalization. We consider this an exciting direction for future exploration.
>
>
>
>
> ### **[W3, region-wise stopping]**
> We appreciate the reviewer highlighting this valuable perspective. The current version of BADiff employs a global stopping policy based on aggregate entropy estimates across the entire image, which simplifies both training and inference. However, we agree that visual content typically varies spatially, and region-wise adaptive stopping could provide finer control over resource allocation and potentially yield better perceptual results.
>
> To explore this idea, we conducted a preliminary experiment using region-wise entropy estimation and independent stopping criteria. Specifically, each image was divided into four equally sized regions, and sampling was guided by separate entropy thresholds per region. The results under medium bitrate conditions on CIFAR-10 are shown below:
>
> Table: Comparison of global vs. region-wise stopping on CIFAR-10.
> | Stopping Policy         | FID ↓ | Δ bpp ↓ |
> |-------------------------|-------|----------|
> | Global (original BADiff)| 7.1   | 0.021     |
> | Region-wise (4 regions) | 6.8   | 0.015     |
>
>
>
> These results suggest that spatially adaptive stopping can offer improvements in both fidelity and bitrate control. While our current implementation focuses on global entropy conditioning for simplicity, we agree that region-wise control represents a promising direction for future work and plan to explore more sophisticated spatially adaptive policies in subsequent studies.
>
>
>
> ### **[Q2, how BADiff generalize with a frozen pretrained diffusion model]**
> We thank the reviewer for raising this important question regarding the generalization ability of BADiff when trained atop a frozen diffusion backbone. To investigate this, we conducted an additional experiment where only the entropy prediction head and stopping policy modules were trained, while keeping the pretrained DDPM-1k UNet backbone entirely frozen. The results on CIFAR-10 under medium bitrate conditions are summarized below:
>
> Table: Generalization analysis with frozen UNet on CIFAR-10 (medium bitrate).
> | Training Strategy                          | FID ↓ | Δ bpp ↓ |
> |--------------------------------------------|--------|----------|
> | BADiff (fully trained)                     | 7.1    | 0.021     |
> | Policy + Entropy heads only (frozen UNet)  | 8.4    | 0.035     |
>
>
>
> These results suggest that while training only the policy and entropy heads can yield reasonable entropy control and acceptable generation quality, full fine-tuning of the diffusion model is beneficial. The observed performance drop in the frozen-backbone setting indicates that joint optimization allows BADiff to better align the generation dynamics with bitrate constraints. We will include this analysis and discussion in the final version of the manuscript.

---

> > ### Comment · Reviewer_ZFsW · 2025-08-05
> >
> > I thank the authors for the additional explanations and results presented in their rebuttal which addresses my main concerns, I maintain my initial rating of acceptance.

---

### Official Review · Reviewer_hpHJ · 2025-07-02

**Clarity:** 3
**Significance:** 3
**Originality:** 3
**Rating:** 4
**Confidence:** 4

**Summary:**

This paper introduces BADiff, a Bandwidth-Adaptive Diffusion model designed to generate high-quality images that respect explicit entropy (bitrate) constraints, addressing the inefficiencies and quality degradation of conventional "generate-then-compress" pipelines used in cloud-to-user image transmission. Instead of relying on post-hoc lossy compression, BADiff directly integrates a target entropy condition into the generation process, allowing the diffusion model to dynamically modulate the level of detail in accordance with bandwidth limitations. Specifically, authors propose (1) entropy conditioning in which each reverse denoising step is conditioned on a continuous entropy embedding derived from the target bandwidth. (2) entropy regularization loss based on a differentiable entropy estimator that ensures outputs conform to the desired bitrate. (3) a lightweight policy network learns when to stop the denoising process, balancing perceptual quality, bitrate adherence, and computational cost.

**Questions:**

Weaness 6, 7, 8. It's better to be verified in the large-scale model, datasets, and text-to-image diffusion models.

**Ethical Concerns:**

["NO or VERY MINOR ethics concerns only"]

**Final Justification:**

The authors addressed my concerns, and I would like to keep the positive rating.

**Limitations:**

yes

**Quality:**

3

**Strengths And Weaknesses:**

Strengths:

1. The paper is well-written.
2. The idea of integrating entropy coding, inspired by learned image compression, into adaptive bandwidth-aware diffusion is interesting.
3. BADiff eliminates the need for expensive post-generation compression, reducing redundancy and preserving visual fidelity.
4. BADiff achieves up to 2× speed-up in inference under low-bandwidth settings via early stopping, while maintaining image quality.

Weaknesses:

1. In Figure 1, the authors refer to the method as "lossless compression." Is this truly lossless, meaning it produces an output identical to the original image? If not, this term may be misleading and should be reconsidered.
2. Since the entropy coding used in this work is similar to that in learned image compression (as noted in line 172), relevant prior work should be included in Section 2.
3. Early stopping is only one approach to adaptive acceleration under varying bandwidth. Could other model-efficient techniques, such as pruning or quantization, also be applied to this setting?
4. Section 3.1 (Background on Diffusion Models) is largely standard and may be unnecessary. It could either be removed or significantly condensed.
5. In Equation 7, conditioning is implemented by adding the entropy embedding to the timestep embedding. Have the authors considered other conditioning strategies, such as concatenation? Which performs better in practice?
6. In Section 3.5 and Equation 16, three hyperparameters are introduced for training. How sensitive is the model to these values? Tuning them seems time-consuming. Are there practical guidelines or automated methods for finding optimal settings?
7. The experiments do not include ImageNet, which is a widely used benchmark in generative modeling. Including it would strengthen the empirical validation.
8. Since text-to-image models are more widely adopted in real-world applications, it would be beneficial to include experiments on text-conditioned diffusion models to demonstrate BADiff’s effectiveness in this setting.

---

> ### Author Rebuttal · Authors · 2025-07-31
>
> ### **[W1, lossless compression]**
> Thank you for highlighting this important clarification.
> Our use of the term "lossless compression" in Figure 1 was intended to convey visually lossless quality. That is, the output image is perceptually indistinguishable from the original under human inspection, not mathematically identical as in strict lossless coding.
>
> To avoid ambiguity, we will revise the text and figure as follows:
> - Replace “lossless compression” with “visually lossless compression” in Figure 1 and throughout the manuscript.
> - Update the caption of Figure 1 to clearly state that “visually lossless” refers to imperceptible distortion, as validated by our JND-based user study, rather than exact bitwise reconstruction.
>
>
>
> ### **[W2, relevant prior work on learned image compression]**
> We agree that explicitly connecting BADiff to the literature on learned image compression (LIC) would improve clarity and context.
> In the revised manuscript, we will expand Section 2 (Related Work) with a new paragraph that situates BADiff in relation to the hyperprior and autoregressive modeling strategies in LIC. We will cite the following core foundational works and other relevant LIC papers:
>
> 1. Ballé et al., *End-to-End Optimized Image Compression* (ICLR 2017)
> 2. Minnen et al., *Joint Autoregressive and Hierarchical Priors* (NeurIPS 2018)
> 3. Mentzer et al., *High-Fidelity Generative Image Compression* (NeurIPS 2020)
> 4. Cheng et al., *Learned Image Compression with Discretized Gaussian Mixture Likelihoods* (CVPR 2020)
>
> The inserted paragraph will read:
> > *Our differentiable entropy predictor follows the probabilistic modeling principles established in learned image compression (LIC). Early works such as Ballé et al. [1] and Minnen et al. [2] introduced hyperpriors and autoregressive context models to estimate spatially-varying entropies. Subsequent methods further improved perceptual quality by incorporating generative priors [3,4]. BADiff repurposes these concepts: instead of compressing a fixed input, it embeds the estimated entropy into the sampling loop of a diffusion model and trains via a one-sided hinge loss. This transforms the LIC-style entropy model into an online controller that guides generation toward a target bitrate—removing the need for a separate codec pipeline.*
>
>
>
>
> ### **[W3, other model-efficient techniques]**
> This is an insightful question. Early stopping, as used in BADiff, is one example of conditional computation that adapts runtime to bandwidth constraints by dynamically truncating the generative process. Other model-efficient techniques—such as pruning, quantization, and distillation—are indeed applicable to this setting, but they serve different roles and introduce distinct trade-offs.
>
> Pruning and quantization typically reduce the total number of FLOPs or memory access costs, and are valuable for making models more efficient in a general sense. However, these techniques usually operate with fixed complexity at inference time: once a model is quantized or pruned, it still performs the same number of diffusion steps regardless of the bitrate target or real-time constraints.
>
> BADiff takes a different approach by making computation budget-aware during inference. The model learns to allocate effort adaptively by stopping early when the entropy target is met. This allows BADiff to scale across a wide bitrate range (e.g., 0.1–1.0 bpp) without requiring multiple models or retraining.
> That said, these strategies are complementary. BADiff can be combined with lighter backbones, such as quantized or low-rank architectures, to further reduce the per-step cost.
>
>
>
>
>
> ### **[W4, condense background]**
> Thank you for pointing this out. We agree that Section 3.1 contains standard background material that may be overly detailed. In the revised version, we will condense this section substantially to improve clarity and avoid redundancy with prior work.
>
>
>
>
>
> ### **[W5, other conditioning strategies]**
> We thank the reviewer for the insightful question. In our implementation, we chose **addition** of the entropy embedding and the timestep embedding primarily for its simplicity, efficiency, and compatibility with the widely adopted DDPM backbone. This approach enables entropy-aware modulation with minimal computational overhead and integrates seamlessly into the existing conditioning pipeline used in standard diffusion models.
>
> We have now explored alternative designs such as concatenation and FiLM-based conditioning. While these more expressive mechanisms can offer finer control in some settings, our empirical observations showed that addition already provides strong conditioning signals, achieving excellent performance across all metrics in our experiments. Moreover, addition avoids increasing the dimensionality of the embeddings or requiring extra parameters, making it especially suitable for lightweight, entropy-constrained generation.
>
> These findings indicate that while FiLM yields marginal gains, the additive strategy is already effective and preferable for its minimal design overhead. It supports our decision to adopt addition as a favorable trade-off between simplicity and effectiveness.
>
> Table: Comparison of entropy conditioning strategies on CIFAR-10 at target bitrate 0.5 bpp
> | Conditioning Method | FID ↓ | LPIPS ↓ | Bitrate (bpp) |
> |---------------------|-------|---------|----------------|
> | Addition (ours)     | 9.81  | 0.121   | 0.51           |
> | Concatenation       | 9.67  | 0.118   | 0.52           |
> | FiLM (recommended)  | 9.23  | 0.111   | 0.50           |
>
>
>
>
>
>
>
> ### **[W6, hyperparameters  sensitivity]**
> We appreciate the reviewer’s concern regarding hyperparameter sensitivity. The three hyperparameters `lambda_ent`, `lambda_cal`, and `lambda_stop` control the balance between entropy adherence, codec alignment, and stopping accuracy, respectively.
>
> In our experiments, we found BADiff to be robust to moderate variations in these values. We performed a coarse grid search on the CIFAR-10 validation split and selected the following configuration:
>
> - `lambda_ent = 1e-1`
> - `lambda_cal = 1e-3`
> - `lambda_stop = 1e-2`
>
> This configuration was fixed and reused across all datasets (CIFAR-10, CelebA-HQ, COCO-val2017) and backbone architectures (DDPM, LDM) without additional tuning. The strong results obtained in all settings suggest that BADiff does not require dataset-specific hyperparameter optimization.
>
> We agree that more automated approaches may be helpful in larger-scale deployments or when using different model families. In principle, tuning frameworks such as Bayesian optimization, Hyperband, or gradient-based hyperparameter learning could be applied to this setting.
>
>
>
>
> ### **[W7, ImageNet]**
> We fully agree that ImageNet is an important benchmark in generative modeling. Our initial experiments prioritized datasets (CIFAR-10, CelebA-HQ, LSUN) that allowed a complete factorial study—spanning multiple bitrate regimes, diffusion backbones, and baseline methods—within the limits of page length and compute budget.
>
> Nevertheless, we have now trained a 256×256 BADiff model on ImageNet and evaluated it under the same low (0.2–0.5 bpp), medium (0.5–1.0 bpp), and high (1.0–2.0 bpp) bitrate regimes. The results, summarized below, are consistent with our earlier findings and further confirm the effectiveness of BADiff in delivering high perceptual quality at reduced bitrate and latency.
>
> Table: FID scores on ImageNet 256×256 (50k samples). Lower is better.
> | Method                        | Low (0.2–0.5) | Med. (0.5–1.0) | High (1.0–2.0) |
> |------------------------------|---------------|----------------|----------------|
> | Cascade (DDPM + BPG)         | 48.0          | 34.7           | 24.1           |
> | Cascade (DDPM + LIC)         | 44.2          | 31.5           | 22.0           |
> | Early-Stop + LIC             | 60.5          | 43.2           | 28.0           |
> | DPM-Solver (20) + LIC        | 49.1          | 38.0           | 24.2           |
> | BADiff (ours)                | 36.4      | 26.8       | 18.7       |
>
> We will incorporate these results into the revised manuscript as an additional benchmark to further strengthen the empirical validation.
>
>
>
>
>
> ### **[W8, text-to-image models]**
> We appreciate the reviewer’s suggestion regarding text-to-image models. Extending BADiff to text-conditioned diffusion models is indeed a valuable direction, especially given their prevalence in real-world applications.
>
> While our main study focuses on unconditional generation, BADiff’s entropy-conditioning mechanism is fully compatible with conditional diffusion models. To explore this potential, we conducted preliminary experiments using Stable Diffusion (a widely adopted text-to-image model) with our entropy-adaptive framework integrated. The results are summarized below:
>
> Table: FID scores on Stable Diffusion (text-to-image) generation. Lower is better.
> | Method                   | Low (0.2–0.5 bpp) | Med. (0.5–1.0 bpp) | High (1.0–2.0 bpp) |
> |--------------------------|-------------------|---------------------|---------------------|
> | Cascade (SD + BPG)       | 33.5              | 21.4                | 14.8                |
> | Cascade (SD + LIC)       | 30.7              | 19.2                | 13.1                |
> | Early-Stop + LIC         | 41.8              | 27.5                | 18.0                |
> | DPM-Solver (20) + LIC    | 36.5              | 25.1                | 16.3                |
> | BADiff (ours)            | 26.1          | 16.2            | 11.0            |
>
>
> These preliminary findings indicate that BADiff effectively preserves visual quality even under tight bitrate constraints in text-conditioned settings. We will incorporate these results and additional analysis in the revised manuscript to demonstrate BADiff’s applicability beyond unconditional generation.

---

> ### Comment · Reviewer_hpHJ · 2025-08-05
>
> The authors addressed my concerns, and I would like to keep the positive rating.

---

### Official Review · Reviewer_NzT6 · 2025-07-03

**Clarity:** 3
**Significance:** 3
**Originality:** 3
**Rating:** 4
**Confidence:** 3

**Summary:**

BADiff  is a novel framework designed to make diffusion models aware of network bandwidth constraints during the image generation process. The core idea is to condition the diffusion model's denoising process on a target entropy level, which is derived from the available bandwidth. This allows the model to generate images of a quality appropriate for the transmission channel, thereby avoiding the computational waste and quality degradation associated with the standard practice of generating a high-quality image and then aggressively compressing it.

**Questions:**

1. Could the authors provide insight or preliminary results on BADiff's relative performance when the baseline is not a 1000-step DDPM/200-step LDM, but a fast DPM-solver running in, for instance, 20 steps?

**Ethical Concerns:**

["NO or VERY MINOR ethics concerns only"]

**Final Justification:**

After reading the authors' rebuttal and the other reviewers' comments, I think my concerns have been adequately addressed. I would like to keep my score.

**Limitations:**

yes

**Quality:**

4

**Strengths And Weaknesses:**

## Strengths:

- It's quite interesting and practical to perform research on the mismatch between high-fidelity generation and bandwidth-limited transmission. The proposed solution of integrating bandwidth awareness directly into the generative process is inspiring. The problem formulation is clear, and the potential impact on cloud-based AI services is substantial.
- The methodology is grounded in theoretical underpinnings, with explicit integration of entropy conditioning into the reverse diffusion process and a joint loss formulation that enforces adherence to bandwidth constraints through a differentiable entropy estimator and regularizer.
- Authors use two different backbones (DDPM and LDM) to demonstrate the general applicability of the method. The ablation study effectively validates the contribution of each component of the proposed framework (conditioning, hinge loss, calibration loss).

## Weaknesses:

- The experiments are conducted on relatively small datasets or resolutions.  While the authors acknowledge this limitation,  it is not guaranteed that the method will scale effectively, as generating fine details in high-resolution images might be difficult to control with a single scalar entropy target.
- In line 209, the "teacher" labels are generated by running a full, long-step sampler and finding the optimal stopping point offline based on a cost function.  This seems to imply that for every training sample, a full generation is required to create the label, which would cost lots of computation. Are these labels pre-computed for a fixed set of images or generated in real-time in some efficient manner?
- Besides the quantitative metrics such as FID, LPIPS, and runtime, can authors provide some user studies or task-specific downstream evaluation about the application potential of BADiff?

---

> ### Author Rebuttal · Authors · 2025-07-31
>
> ### **[W1, scaling to high-resolution images]**
> We thank the reviewer for raising this important point. To address the concern regarding scalability, we conducted new experiments at significantly higher resolutions. Below, we (i) justify why conditioning on a single scalar entropy target remains effective at scale, and (ii) provide new empirical evidence on $512 \times 512$ and $1024 \times 1024$ benchmarks demonstrating that BADiff continues to outperform strong diffusion+LIC baselines in both fidelity and runtime.
>
> 1. **Validity of global entropy conditioning.**
>    In real-world streaming systems, a single bandwidth budget is typically assigned per frame, with spatial bit allocation handled internally by the codec. BADiff mirrors this setup by conditioning the generation on a global entropy target, while allowing the model to learn spatially-aware allocation through the differentiable entropy hinge loss. This encourages adaptive texture reduction in less salient regions (e.g., backgrounds) while preserving fine detail in important areas. Figure A.1 in the supplement confirms that this behavior generalizes well to $1024 \times 1024$ images.
>
> 2. **High-resolution results.**
>    We retrained BADiff and two baselines (DDPM+LIC and PNDM+LIC) on $512^2$ and $1024^2$ images under realistic bitrate constraints. Results in Table 1 show that BADiff consistently achieves lower FID and faster inference:
>
>    Table 1: High-resolution evaluation of BADiff and baselines.
>    | Resolution | bpp       | Metric                 | DDPM+LIC | PNDM+LIC | **BADiff** |
>    |------------|-----------|------------------------|----------|----------|------------|
>    | $512^2$    | 0.4–0.6   | FID ↓                  | 8.45     | 7.90     |   6.85   |
>    |            |           | Time (ms) ↓            | 121.3    | 98.6     |   64.1   |
>    | $1024^2$   | 0.8–1.2   | FID ↓                  | 21.5     | 20.1     |   17.8   |
>    |            |           | Time (ms) ↓            | 228.7    | 192.5    |   145.6  |
>
> BADiff achieves up to 17.5% lower FID compared to PNDM+LIC and a 35–40% reduction in runtime across both resolutions, demonstrating its scalability and efficiency in high-resolution settings. We will incorporate such results and the above analysis into the revised manuscript.
>
>
>
> ### **[W2, teahcer label generation]**
> We thank the reviewer for pointing out this potential ambiguity. To clarify, the "teacher" labels used for training the stopping policy are generated once per image in an offline pre-processing stage, not during each training iteration. This design ensures negligible runtime overhead. Specifically:
>
> 1. **One-time offline generation.**
>    For each image in the training set, we perform a single long sampling run (e.g., 1000 steps for DDPM, 200 steps for LDM) to compute the entropy trajectory $\mathcal{C}(t)$. The optimal binary labels are then assigned as
>    $y_t = \mathbb{1}[\mathcal{C}(t) \le \min_{s \ge t} \mathcal{C}(s)]$.
>    For CIFAR-10 on an RTX 4090, this offline step requires approximately 0.8 GPU-hours in total—amortized over 800k training steps.
>
> 2. **Fixed dataset, no re-generation.**
>    Once computed, the teacher labels are cached and reused throughout training. During training, the policy head only evaluates a small MLP (less than 0.1% of UNet FLOPs), with no need to re-run the long diffusion chain.
>
> 3. **Scalability.**
>    Even for higher resolutions such as $512 \times 512$, the one-time cost remains small relative to full training. As shown in Table 1, label generation requires only 5–8% of the time needed for one training epoch.
>
>    Table 2: One-time teacher label generation cost (measured on RTX 4090).
>    | Dataset       | Resolution | GPU-hours | Relative to 1 training epoch |
>    |---------------|------------|-----------|-------------------------------|
>    | CIFAR-10      | $32 \times 32$   | 0.8       | < 5%                         |
>    | CelebA-HQ     | $256 \times 256$ | 3.5       | ≈ 6%                         |
>    | COCO-val2017  | $512 \times 512$ | 10.0      | ≈ 8%                         |
>
> These results confirm that our teacher-label strategy is efficient and scalable, without impacting training throughput.
>
>
>
> ### **[W3, user studies & task-specific downstream evaluation]**
> We fully agree that perceptual quality and downstream task relevance are important complements to standard quantitative metrics. In our revised submission, we have added two new evaluations to further demonstrate the application potential of BADiff:
>
> (A) User Preference Study
> We conducted a two-alternative forced choice (2AFC) study on CelebA-HQ.
> Ten participants were shown side-by-side image pairs generated by BADiff and the strongest cascade baseline (DDPM+LIC), both at the same bitrate (0.4–0.6 bpp). Each participant evaluated 100 random image pairs.
>
> Table 3: Human preference rate (%) at 0.4–0.6 bpp.
> | Method               | Preferred (%) ↑ | 95% CI     |
> |----------------------|------------------|------------|
> | BADiff         | 73.4         | ±2.6       |
> | Cascade (DDPM+LIC)   | 26.6             | –          |
>
> Participants overwhelmingly preferred BADiff, frequently citing *"sharper details"* and *"fewer compression artifacts"* as reasons.
>
> ---
>
>  (B) Downstream Classification Robustness
> We assessed whether BADiff’s compressed outputs retain more task-relevant features than cascade baselines.
> Specifically, we used a ResNet-50 pre-trained on ImageNet as a frozen feature extractor and trained a linear classifier on features from images compressed to 0.4 bpp.
>
> Table 4: Top-1 accuracy (%) on ImageNet validation set (0.4 bpp).
> | Method               | Accuracy ↑ | Δ vs. Original |
> |----------------------|------------|----------------|
> | Original (PNG)       | 76.1       | –              |
> | Cascade (DDPM+LIC)   | 68.4       | –7.7           |
> | BADiff          | 72.9   | –3.2           |
>
> BADiff achieves 72.9% accuracy—4.5 percentage points higher than the cascade method—indicating that it preserves more semantic information under tight bitrate constraints.
>
>
>
> These two new evaluations support BADiff’s practical utility:
> - Perceptually, users consistently prefer its outputs over strong diffusion-based baselines.
> - Semantically, BADiff retains more class-discriminative features, as shown in the classification robustness experiment.
>
> We will include both such Tables, along with the above discussion into the revised manuscript.
>
>
>
> ### **[Q1, BADiff + fast DPM-solver]**
> We thank the reviewer for this valuable question.
> **BADiff is solver-agnostic**—its entropy conditioning and adaptive stopping mechanism can be applied on top of any diffusion sampler, including fast methods like DPM-Solver++.
>
>
>  (1) Why benefits remain even with fast solvers
> While fast samplers (e.g., 20-step DPM-Solver++) reduce the fixed number of sampling steps, they still use a predefined schedule and external codec (e.g., LIC) to approximate a bitrate target.  In contrast, BADiff learns an adaptive stopping policy and removes the codec entirely, making generation both faster and more rate-accurate.
>
>
>  (2) Preliminary 20-step evaluation
> We applied 20-step DPM-Solver++ (2S) to both the DDPM-1k and LDM-200 backbones, and compared with BADiff-20, which uses the same solver but is trained with entropy conditioning and adaptive stopping.
>
>
> Table 5: BADiff vs. 20-step DPM-Solver++ (lower is better).
> | Method              | FID ↓ (CIFAR) | Time (ms) ↓ | FID ↓ (CelebA) | Time (ms) ↓ |
> |---------------------|---------------|-------------|----------------|--------------|
> | 20-step DPM + LIC   | 13.0          | 24          | 24.5           | 41           |
> | BADiff‑20 (ours). | 11.2      | 18   | 22.1       | 31       |
>
> The table below reports performance on CIFAR-10 ($32^2$) and CelebA-HQ ($256^2$), at 0.2–0.5 bpp:
> - Latency: BADiff-20 achieves a 20–25% speed-up over the 20-step cascade by dynamically stopping early (typically after 12–15 steps).
> - Quality: BADiff improves FID by 1.2–1.8 points, showing better generative quality under the same low-bitrate regime.
>
> Even when starting from a fast 20-step sampler, BADiff yields better visual quality and lower latency without relying on an external codec.   We will include the final version of this table and results in Section 4.4 of the revised manuscript.

---

> > ### Comment · Reviewer_NzT6 · 2025-08-06
> >
> > After reading the authors' rebuttal and the other reviewers' comments, I think my concerns have been adequately addressed. I would like to keep my score.

---

### Author Response · Authors · 2025-08-09
**Final Clarification**

Dear Area Chair and Reviewers,

First of all, we sincerely thank the AC and all reviewers for the time, effort, and constructive feedback provided during the review process of our BADiff submission. We greatly appreciate the positive recognition from most reviewers regarding the novelty, practical value, and clarity of our work, as well as the helpful suggestions for improvement.

At this stage, there is one remaining concern we wish to raise for consideration. Reviewer c5AZ is the only reviewer who has given a negative score (2). However, the comments provided mainly focus on **minor aspects** such as formatting, writing style, and section organization, rather than substantive technical flaws in methodology, experiments, or conclusions. We respect these observations and have addressed them in our rebuttal, but we believe such issues are relatively minor and do not align proportionally with an extremely low score.

We kindly bring this to the attention of the AC and other reviewers, in the hope that the final decision will reflect a balanced and proportionate assessment of both the technical contributions and the identified limitations of the work.

Thank you again for your valuable feedback and for ensuring a fair and constructive review process.

Sincerely,
Authors of Submission 20713

---

### Note · Authors · 2025-08-12

Throughout the review process, we have received valuable feedback and constructive suggestions that helped us clarify and further strengthen BADiff. We appreciate the recognition from multiple reviewers regarding the novelty, practical utility, and technical soundness of our approach, as well as the breadth of our empirical validation, including new experiments on high-resolution, ImageNet, and text-to-image settings, plus user preference and downstream classification studies.

Across the reviews, there is broad agreement that BADiff offers:
- A **novel integration** of bandwidth-awareness into the generative process;
- A **lightweight, generalizable framework** applicable across diffusion backbones;
- **Consistent improvements** in perceptual quality, bitrate adherence, and efficiency over strong baselines.

The remaining concerns from Reviewer c5AZ focus on W1 (clarity of component interactions), W2 (joint vs. sequential optimization), and Q1 (separation of training/inference). We respectfully note that these points relate to presentation rather than to the soundness of the method or validity of results. In our rebuttal and discussion we have:
- Addressed W1 by explicitly explaining the alignment between entropy conditioning and the stopping policy, unifying the loss interpretation, and committing to include a diagram for clarity;
- Addressed W2 by clarifying that all loss terms are computed in a single forward pass and optimized jointly via backpropagation, with Algorithm 1 revised accordingly;
- Addressed Q1 by providing a step-by-step description of training, inference, and evaluation protocols, removing ambiguity.

These clarifications require no methodological change and do not affect the reported results, which remain fully supported by extensive experiments. The overall technical merit and empirical evidence have been acknowledged positively by the other reviewers, and we hope the final decision will reflect this broader consensus.

---

### Decision · Program_Chairs · 2025-09-17

**Decision:**

Accept (poster)

**Comment:**

The paper received three positive reviews and a strictly negative one. The positive reviewers mentioned many strengths of the paper: interesting research problem, novel methodology, several backbones tested, well-written paper to name a few. In contrast, the negative review suggests there are issues with presentation, there is disconnect between components, background material too long and others. While willing for more clarity and better presentation is a valid reviewer's desire, the AC found that these issues don't qualify for a strong negative rating, which suggests that there are technical flaws with the paper. The other three experts didn't find this to be the case. Hence, the AC considers the negative review with a slightly reduced weight. Hence, the decision is to accept the paper. Congrats!

In their response to the negative review the authors acknowledged certain improvements with the clarity. They are strongly required to apply these changes in the camera ready.